# The quantitative metabolome is shaped by abiotic constraints

Amir Akbari [1✉], James T. Yurkovich [1,2], Daniel C. Zielinski[1] & Bernhard O. Palsson [1,3✉]

Living systems formed and evolved under constraints that govern their interactions with the inorganic world. These interactions are definable using basic physico-chemical principles. Here, we formulate a comprehensive set of ten governing abiotic constraints that define possible quantitative metabolomes. We apply these constraints to a metabolic network of *Escherichia coli* that represents 90% of its metabolome. We show that the quantitative metabolomes allowed by the abiotic constraints are consistent with metabolomic and isotope-labeling data. We find that: (i) abiotic constraints drive the evolution of high-affinity phosphate transporters; (ii) Charge-, hydrogen- and magnesium-related constraints underlie transcriptional regulatory responses to osmotic stress; and (iii) hydrogen-ion and charge imbalance underlie transcriptional regulatory responses to acid stress. Thus, quantifying the constraints that the inorganic world imposes on living systems provides insights into their key characteristics, helps understand the outcomes of evolutionary adaptation, and should be considered as a fundamental part of theoretical biology and for understanding the constraints on evolution.

[1] Department of Bioengineering, University of California San Diego, La Jolla, CA, USA. [2] Institute for Systems Biology, Seattle, WA, USA. [3] Novo Nordisk Foundation Center for Biosustainability, Technical University of Denmark, Lyngby, Denmark. ✉email: amakbari@ucsd.edu; palsson@ucsd.edu

Life developed under a variety of environmental pressures, such as pH, temperature, and osmotic pressure that have driven the evolution of robust mechanisms of cellular adaptation[1,2]. Through variations in metabolite levels, metabolic networks sense, adapt to, and influence the local environment to maintain functioning metabolic pathways under diverse and unpredictable conditions[3–5]. We have not yet achieved a quantitative understanding of how various environmental variables influence metabolite levels.

Constraint-based modeling provides a quantitative framework to study the interactions between cellular metabolism and the environment[6]. Constraint-based modeling of reaction fluxes, through flux-balance analysis (FBA), has been successful in predicting the growth rate and metabolic phenotypes by imposing mass-balance constraints on the flow of metabolites through the network[7,8]. Constraint-based formulation of metabolite levels, based instead on thermodynamic constraints, has also been shown to be feasible[9,10]. Unlike FBA, many more biophysical considerations must be accounted for when specifying the constraints on the quantitative metabolome. Moreover, while maximizing the growth rate is an intuitive objective to determine optimal pathway usage, cellular objectives that may underlie the quantitative metabolome have not yet been identified.

In this article, we introduce a comprehensive set of abiotic constraints (ABCs) on the metabolome. We present an ABC-based analysis to determine bounds on metabolite concentrations for a prescribed environment. We validate the results for a metabolic network of *Escherichia coli* against measured concentrations and known thermodynamic bottlenecks of central-carbon pathways. We then use the ABC-based analysis to evaluate how intracellular concentrations are influenced by the environment and shape biological functions. The ABCs are universal and apply from primitive to modern cells across the tree of life. Thus, this work paves the way to biophysically accurate representations of the quantitative metabolome at scales, where one can ask fundamental questions about the evolution and function of cellular processes.

## Results

**Formulation of the ABC-based analysis.** We introduce an ABC-based approach to characterize the constraints that govern the interactions of biological functions with the inorganic world. This approach encompasses ten fundamental and evolutionary classes of abiotic constraints on cellular functions (see Fig. 1 and "Abiotic constraints" section): (i) thermodynamics, (ii) charge balance, (iii) solubility, (iv) membrane potential, (v) buffer capacity, (vi) enzyme saturation, (vii) ionic strength, (viii) osmotic balance, (ix) ion binding, and (x) experimentally unresolved metabolites (Fig. 1a).

Mathematically, thermodynamic and solubility constraints are represented by inequalities and the rest by equalities. Together, these ten classes of constraints define a nonconvex and possibly disconnected set of feasible concentration states that is challenging to characterize (Fig. 1b).

The conceptual basis for the ABC-based analysis is analogous to the widely used flux-balance analysis[8]. A biologically relevant concentration solution space (CSS) is constructed by restricting the concentrations to those obeying all the ABCs. Thus, the CSS contains all quantitative metabolomes that satisfy the ABCs. Accordingly, the ABC-based analysis can be leveraged to determine bounds on metabolite concentrations and Gibbs free energy of reactions.

An objective function may be used to ascertain a particular concentration state from the CSS. We study two cellular objectives pertaining to the concentration state of the cell, namely the thermodynamic and enzyme-saturation efficiency. These objectives play the same role in the ABC-based analysis as the growth-rate objective does in FBA[11] (see "Methods" section for detailed mathematical formulations and computational approaches). They both rest on the assumption that intracellular concentrations are regulated to ensure efficient enzyme utilization, regardless of the growth state[12]. To achieve a prescribed flux state at minimum protein cost, the first maximizes the thermodynamic driving force of all reactions in the network independently of enzyme saturation levels, whereas the second maximizes the saturation level of all enzymes independently of thermodynamic driving forces. Within the framework of the ABC-based analysis, how well these objectives could represent the functional state of the cell is determined by identifying the location of metabolomic data inside the CSS.

Characterization of the CSS for genome-scale metabolic models is computationally challenging. Therefore, we applied the ABC-based analysis to a reduced metabolic network of *E. coli*, comprising 78 reactions, 72 cytoplasmic, and 11 periplasmic metabolites (Supplementary Fig. S3). This network consists of (i) high-flux pathways and (ii) pathways connecting the reduced network to key metabolites with the largest concentrations among those reported in the literature[13,14]. Major pathways, such as glycolysis, tricarboxylic acid (TCA) cycle, anaplerotic reactions, electron transport chain (ETC), biosynthetic sugar metabolism, and cofactor interconversions are included in this network. Importantly, this reduced network contains over 90% of the observed metabolome by mole and includes all reactions with experimentally measured fluxes[14]. Therefore, it possesses the most essential characteristics of the global metabolome for the ABC-based analysis.

We present our results in three parts. First, we show that the ABC-based analysis leads to results that are consistent with concentration and flux data. We also elaborate on the implications of our analysis for other constraint-based approaches, such as thermodynamics-based metabolic flux analysis (TMFA)[10]. Then, we show that the ABCs necessitate the evolution of multiple specialized phosphate transporters. Finally, we provide two case studies, illustrating how the ABCs shape transcriptional responses to stress conditions.

**Quantitative metabolomes defined by the ABCs are consistent with experimental data.** We sought to determine whether the ABCs were consistent with available experimental metabolomic data. We considered growth on four carbon sources, including glucose, acetate, pyruvate, and succinate. The flux states of the network were determined using the latest genome-scale metabolic network reconstruction of *E. coli*[15]. The FBA model (see "Flux distribution" section) was solved to identify fluxes that best matched their experimentally measured values[14] for each carbon source. The resulting flux directions were then used to specify the thermodynamic constraints.

To assess the consistency of metabolomic data with the results of the ABC-based analysis, we identified a point inside the CSS, referred to as computed concentrations $C_{cm}$ (Fig. 2b, crosses), that is the closest to the experimental measurements of Gerosa et al.[14] (Fig. 2b, circles) for each carbon source. We found a strong correlation (Pearson coefficient $R = 0.84$–$0.93$) and relatively small root-mean-square deviation (RMSD = 2.3–5.1 mM) between the computed and experimental concentrations across all carbon sources (Fig. 2a), showing that the CSS contains the measured concentrations. Note that, when only the thermodynamic constraints are included in the ABC-based analysis, the computed concentrations exhibit a higher correlation with metabolomic data, reflecting the restrictive characteristic of the

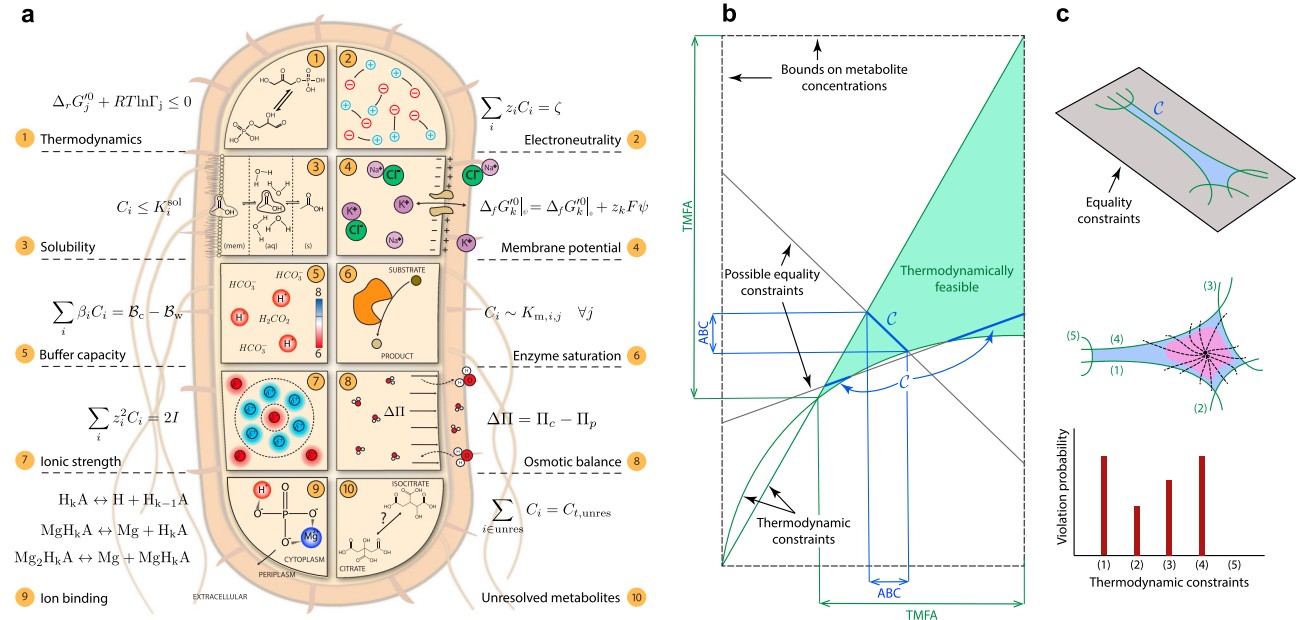

**Fig. 1 Constraint-based formulation of the ABCs determines the concentration state of metabolic networks. a** Schematic representation of ten fundamental and evolutionary constraints that define a concentration solution space $\mathcal{C}$. Reactants, reactions, and species are associated with the indices $i$, $j$, and $k$, respectively. **b** Construction of $\mathcal{C}$ from the ABCs, formulated as equalities (e.g., charge balance) and inequalities (e.g., thermodynamic), illustrated in a two-dimensional space. The solution space $\mathcal{C}$ can be nonconvex or disconnected. Feasible concentration ranges furnished by the ABC-based analysis are generally more restricted than those provided by thermodynamic constrains alone (TMFA analysis[10]). **c** Sampling the solution space by generating random trajectories in $\mathcal{C}$ from an interior point. Interior points are identified in a region (purple area), where experimental data most likely fall. Trajectories are continued until at least one of the thermodynamic constraints is violated. For each thermodynamic constraint, the violation probability is defined as the ratio of the number of trajectories intersecting it and the total number of trajectories.

charge-related constraints of the ABCs (see "Consistency of the ABCs with metabolomic data" section and Supplementary Fig. S12).

We ascertained feasible concentration ranges for all intracellular metabolites in the reduced network when glucose is the sole carbon source (Fig. 2b). The ABC-based analysis reveals a qualitative connection between the network topology and the global concentration bounds. The upper and lower concentration bounds are generally restrictive for hub metabolites (e.g., glucose 6-phosphate and fructose 6-phosphate), while they exhibit the opposite trend for end-node metabolites (e.g., trehalose and UDP-$N$-acetyl-glucosamine). The hub metabolites simultaneously participate in several reactions, and their possible concentrations are constrained to a narrow range, so that they satisfy all the thermodynamic constraints these reactions are subject to. The lower bound for exported metabolites with a low periplasmic concentration is restrictive (e.g., $CO_2$) because the cell must retain a minimum cytoplasmic concentration to sustain a thermodynamic driving force across the cell membrane. For a similar reason, the upper bound for imported metabolites with a low periplasmic concentration is also restrictive (e.g., choline).

To derive order-of-magnitude estimates of cytoplasmic concentrations, we computed expected values for concentrations (Fig. 2b, asterisks) by sampling the CSS (see "Characterization of concentration solution space" section). We found a good agreement between the expected values and measured concentrations for all metabolites, except for a few cofactors, such as ATP, NADP, and NADH (RMSD = 5.47 mM, excluding the cofactors). These cofactors participate in many reactions across the network. They are highly regulated with concentrations spanning several orders of magnitude depending on the growth condition or in response to environmental stresses[10,14]. As a result, their concentrations may be driven towards the CSS boundaries to accommodate a desired flux state. Thus, they cannot be generally well-approximated by the expected values, which are usually interior points of the CSS.

Enzyme-saturation efficiency has been proposed in the literature as a cellular objective to determine optimal intracellular concentrations[12]. To maximize this objective, metabolite concentrations are expected to be of the same order as, or greater than, the Michaelis constants of the reactions the metabolites participate in. However, this heuristic principle does not apply to all enzymes (e.g., degradation reactions)[13]. Nevertheless, it is expected to hold for reactions in central-carbon metabolism, where flux directions alter in response to stress conditions, so that enzymes are efficiently saturated in both directions[13].

We compared the maximum Michaelis constant $K_m^{max}$ (Methods: Eq. (53)) of metabolites (Fig. 2b, dark gray bars) with measured concentrations (Fig. 2b, circles). We performed this comparison only for reactions with known Michaelis constants. We found 14 undersaturated reactions ($C_i < K_{m,i}^{max}$) from central-carbon pathways (e.g., PGI, ICDHyr, TALA, and AKGDH), indicating that saturation efficiency is not a fundamental cellular objective or constraint, with which to determine the concentration state of metabolic networks. This result is consistent with previous studies on the efficiency of enzymatic reactions[13,16]. Nonetheless, saturation efficiency could be one of several competing factors determining the functional state of the cell[17].

**Thermodynamic bottlenecks identified by the ABC-based analysis are consistent with isotope-labeling measurements.** Thermodynamics play a fundamental role in adaptation. Modern bacteria have an extraordinary ability to grow and survive using minimal environmental resources. To maximize their survival chances, they efficiently allocate available resources to essential biochemical reactions, maintaining a steady flow of materials

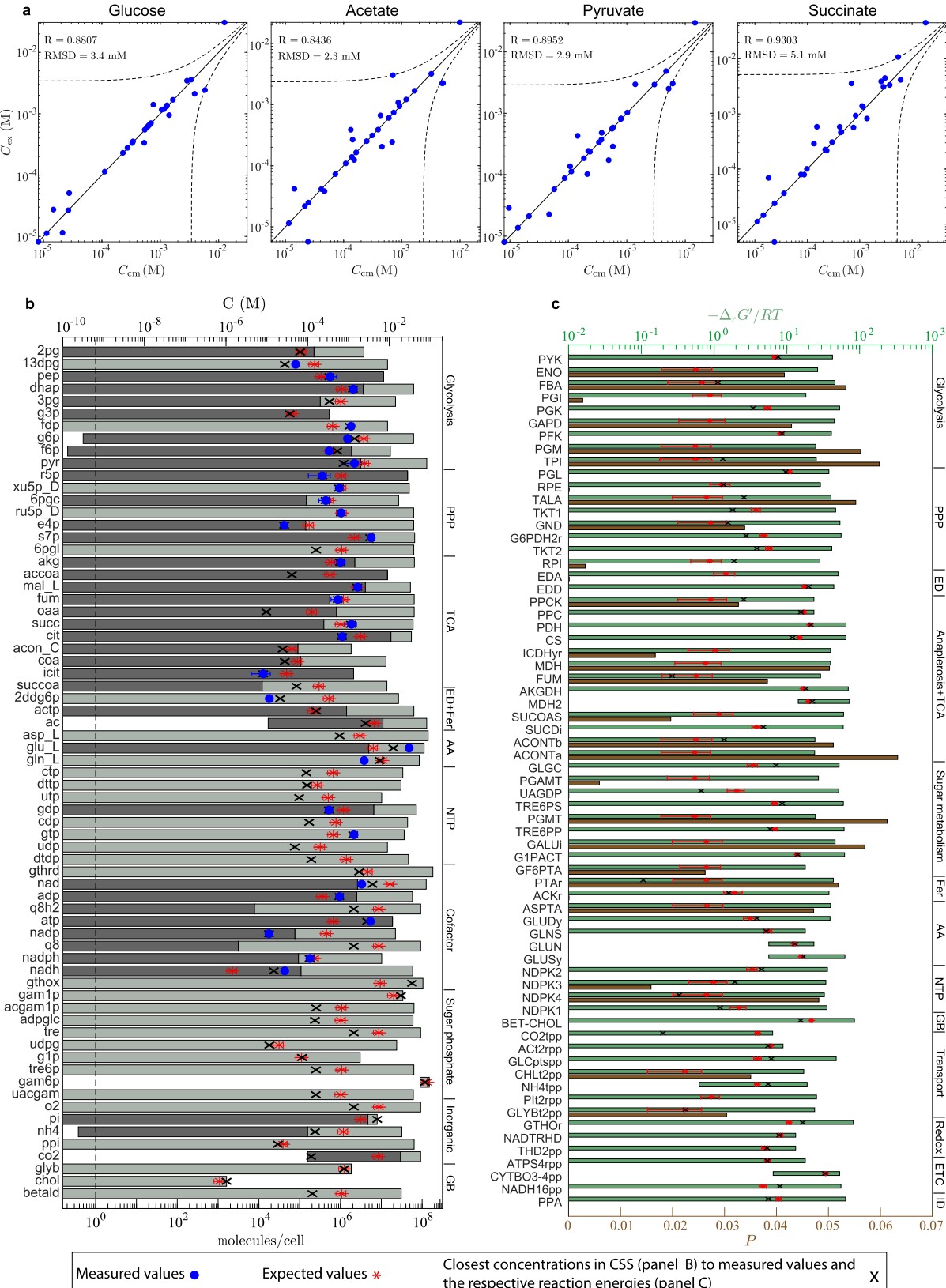

through all their metabolic pathways. Sustaining this optimal flux state hinges on a trade-off between pathway thermodynamics and proteome allocation. Reactions with a small thermodynamic driving force have higher protein cost and vice versa[17]. Through this trade-off, metabolic fluxes are adapted to environmental perturbations[18].

We sought to determine the thermodynamic bottlenecks resulting from the ABCs. We calculated the feasible ranges of transformed Gibbs energy of reactions (Fig. 2c, green bars) and the violation probabilities (see Fig. 1c and its caption for the definition) of their respective thermodynamic constraints (Fig. 2c, brown bars) to identify the most constraining reactions of the

**Fig. 2 The ABCs are consistent with metabolomic and isotope-labeling data for _E. coli_. a** Comparison of measured concentrations $C_{ex}$ and computed concentrations $C_{cm}$. Here, computed concentrations refer to a point inside the CSS that is the closest to the experimental measurements of Gerosa et al.[14]. Dashed lines indicate deviation by root-mean-square deviation (RMSD) from the line $C_{ex} = C_{cm}$. The Pearson correlation coefficient is denoted $R$. **b** Feasible ranges of metabolite concentrations. Dark gray bars, determined from known Michaelis constants[63] (Methods: Eqs. (53) and (54)), show part of the feasible range, where some of the reactions a given metabolite participates in are undersaturated. Measured concentrations are reported by Gerosa et al.[14]. The number of molecules per cell is calculated from the respective molar concentration based on $V_{cell} = 2.6$ fL[64]. **c** Feasible ranges of transformed Gibbs energy of reactions and violation probabilities of thermodynamic constraints. Near-equilibrium reactions are those, for which the scaled transformed Gibbs free energy ($\Delta_r G'/RT$), evaluated at a point inside the CSS that is the closest to experimental values (crosses), is smaller in magnitude than $10^{-2}$. Therefore, the corresponding points are not shown on the respective green bars. For **b** and **c**, glucose is the sole carbon source. Pathway abbreviations are defined in Supplementary Table S3. Error bars indicate twice the standard deviation.

reduced network when glucose is the sole carbon source. Reactions with negative upper and lower bounds on transformed Gibbs energy (e.g., MDH2) are irreversible; all the other reactions can be either inactive or reversible. In FBA, every reaction is considered reversible if no information about Gibbs energies is available. However, using the feasible ranges of transformed Gibbs energy computed from the ABC-based analysis, the flux solution space can be reduced by imposing the corresponding flux bounds. We also found that, if the expectation of transformed Gibbs energy of a reaction is of the same order as $RT$ or less, then the violation probability of that reaction is substantially larger than other reactions, providing a qualitative criterion for when a reaction is thermodynamically constraining in a metabolic network.

Reaction energies and concentrations are restricted more by the ABCs than thermodynamic constraints alone. As a result, the global concentration bounds derived from the ABC-based analysis can either coincide with (e.g., upper bound for choline) or be more restrictive than (e.g., lower bound for glucose 6-phosphate) those furnished by TMFA (Fig. 3a). The thermodynamic constraints are more dominant in the first, and charge-related constraints are more dominant in the second. The dominant constraints set the limit for the metabolite concentrations. We compared the feasible ranges of transformed Gibbs energy of reactions arising from the ABC-based analysis and TMFA (Fig. 3b), identifying five irreversible reactions (GLUN, GLUSy, MDH2, NH4tpp, CYTBO3-4pp) compared to the three provided by TMFA (MDH2, NH4tpp, CYTBO3-4pp).

We illustrated the thermodynamic bottlenecks of the reduced network on a network map (Fig. 4a) to visualize metabolite concentrations and transformed Gibbs energy of reactions. These were evaluated at a point inside the CSS with the minimum distance from the experimental data of Gerosa et al.[14]. We found several reactions in glycolytic (ENO, PGI, GAPD, PGM), TCA cycle (ICDHyr, MDH, ACONTa, SUCOAS), and sugar metabolism (PGMT, PGAMT, GF6PTA, GALUi) pathways with small transformed Gibbs energy of reactions ($|\Delta_r G'| \ll 1$ kJ/mol). The feasible concentration ranges of metabolites participating in these reactions are not necessarily restrictive, so they are regulated to operate close to their equilibrium.

Four major findings from the foregoing results are notable: (i) Near-equilibrium reactions from the ABC-based analysis agree with those previously reported for glycolysis[18] and the TCA cycle[19,20] based on $^2$H and $^{13}$C metabolic flux analysis. (ii) Flux-direction reversal in glycolysis is often induced by carbon-source and nutrient alterations[14]. Thus, having near-equilibrium steps enhances the energy efficiency of pathways, enabling the cell to rapidly switch flux directions in response to energy and biomass demands with minimal concentration changes[18]. (iii) The analysis indicates that there are several thermodynamic bottlenecks in the TCA cycle, hampering its full cyclic operation[19]. (iv) The results suggest that thermodynamic

efficiency (see "Enzyme kinetics" section) is not a fundamental cellular objective that determines the concentration state of metabolic networks.

To quantify the degree to which the additional constraints of the ABC-based analysis can reduce the feasible ranges, we compared the feasible ranges of concentrations and transformed Gibbs energy of reactions furnished by the ABC-based analysis and TMFA when glucose was the only carbon source. We found that the upper bounds on metabolite concentrations and transformed Gibbs energy of reactions from TMFA were on average 728% and 11% larger than those from the ABC-based analysis, and the lower bounds were on average 7% and 87% smaller.

**Thermodynamic constraints drive the evolution of high-affinity phosphate transporters.** By examining individual ABCs in isolation, one can identify the most dominant constraints under a given condition. Such identification may have important evolutionary implications because each of these constraints could have been dominant at different point in time during evolution. For example, thermodynamic laws are believed to have constrained the evolution of biochemical-reaction networks ever since the formation of primitive cells to ensure the energy requirements and spontaneity of prebiotic chemical reactions[21]. However, the osmotic-pressure differential and membrane potential likely have changed significantly during the course of evolution from ion-permeable porous membranes in protocells to ion-impermeable lipid-bilayers in modern cells[22]. This forms a rational basis for chronological postdictions concerning the evolution of alternative pathways.

_E. coli_ has two major phosphate-transport systems (Fig. 3e), namely _Pit_ (low-affinity) and _Pst_ (high-affinity). We sought to determine whether any of the ABCs could have been restrictive enough to hinder the operation of the _Pit_ system in phosphate-limited environment, thereby driving the evolution of the _Pst_ system. These transporters, which differ in their stoichiometry (Fig. 3e), are of particular interest because inorganic phosphate is an important constituent of energy-carrying molecules (e.g., ATP), cofactors (e.g., NADH), and information-storage molecules (e.g., DNA). Therefore, _E. coli_ maintains an optimal homeostatic phosphate level around 1–10 mM[23,24].

We, thus, asked: Is it possible to achieve cytoplasmic-phosphate concentrations above 1 mM while satisfying all the ABCs? We computed feasible concentration ranges for the reduced network, in which either the _Pit_ or _Pst_ system was the only active phosphate transporter. We compared these feasible ranges at two periplasmic concentrations, representing phosphate-limited and phosphate-rich environments (Fig. 3c). While the feasible concentration range for the cytoplasmic phosphate is almost insensitive to its periplasmic concentration when the _Pst_ system is active, achieving cytoplasmic concentrations above 1 mM is impossible in phosphate-limited environments when the _Pit_ system is active. Moreover, comparing the

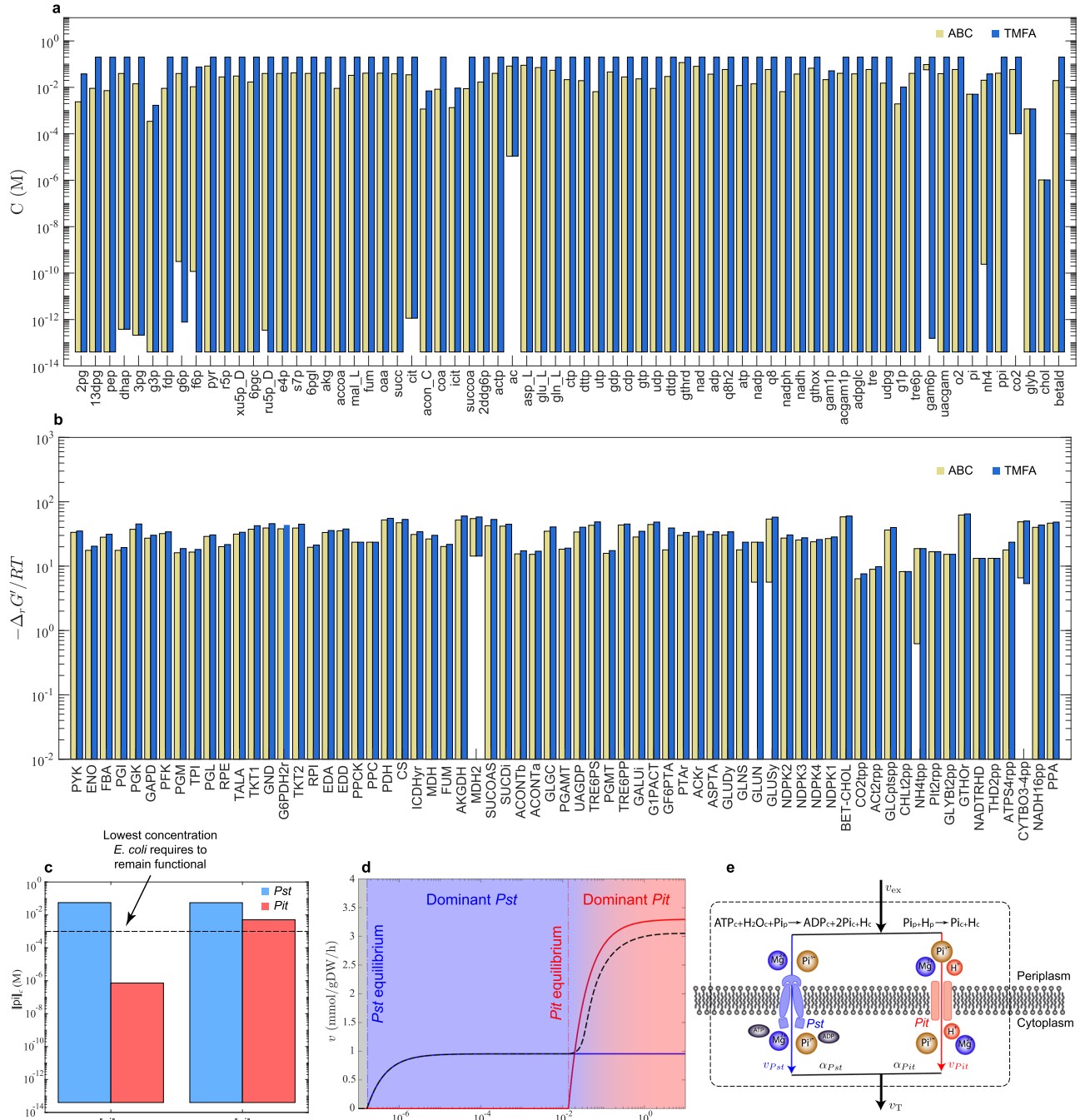

**Fig. 3 Abiotic constraints underlie the evolution of alternative transport systems.** Feasible ranges of metabolite concentrations (**a**) and transformed Gibbs energy of reactions (**b**) furnished by the ABC-based analysis and TMFA are compared. **c** Feasible cytoplasmic-phosphate concentration ranges in mutants, where either the *Pst* system (blue) or the *Pit* system (red) is active at periplasmic concentrations $[pi]_{p,1} = 0.01$ and $[pi]_{p,2} = 69$ mM. **d** Phosphate-transport kinetics in *E. coli*. Solid blue and red lines show phosphate uptake rates in mutants, where either the *Pst* system or the *Pit* system is active, and dashed black line corresponds to a wild-type strain, where both systems are active. Maximum velocities and Michaelis constants are taken from Willsky et al.[65]. The total phosphate uptake rate is estimated $v_T = \alpha_{Pst}v_{Pst} + \alpha_{Pit}v_{Pit}$ with $\alpha_{Pst} = 0.09$ and $\alpha_{Pit} = 0.9$ at large $[pi]_p$.[65] Uptake rates are estimated using $[pi]_c = 1$, $[atp]_c = 3.45$, and $[adp]_c = 0.6$ mM. **e** Schematic representation of the *Pst* and *Pit* systems operating in parallel in wild-type strains.

upper bound on the cytoplasmic-phosphate concentration provided by the ABC-based analysis and TMFA (Fig. 1b) at the foregoing periplasmic concentrations reveals that the operation of the *Pit* system is limited by the thermodynamic constraints, not the osmotic-balance or other charge-related constraints (see "Role of phosphate transporters" section).

Next, we asked: Can high-affinity transporters maintain sufficient phosphate influx in phosphate-limited environment to

support growth? We compared the transport kinetics of the two phosphate transport systems (Fig. 3d) in a wild-type strain of *E. coli* to demonstrate the transition between the *Pit* and *Pst* systems in response to phosphate-starvation stress. Clearly, *E. coli* can remain functional in a wide range of extracellular concentrations, importing phosphate through the *Pit* system at high rates when it is available in excess. However, when the *Pit* system reaches its equilibrium at low extracellular concentrations,

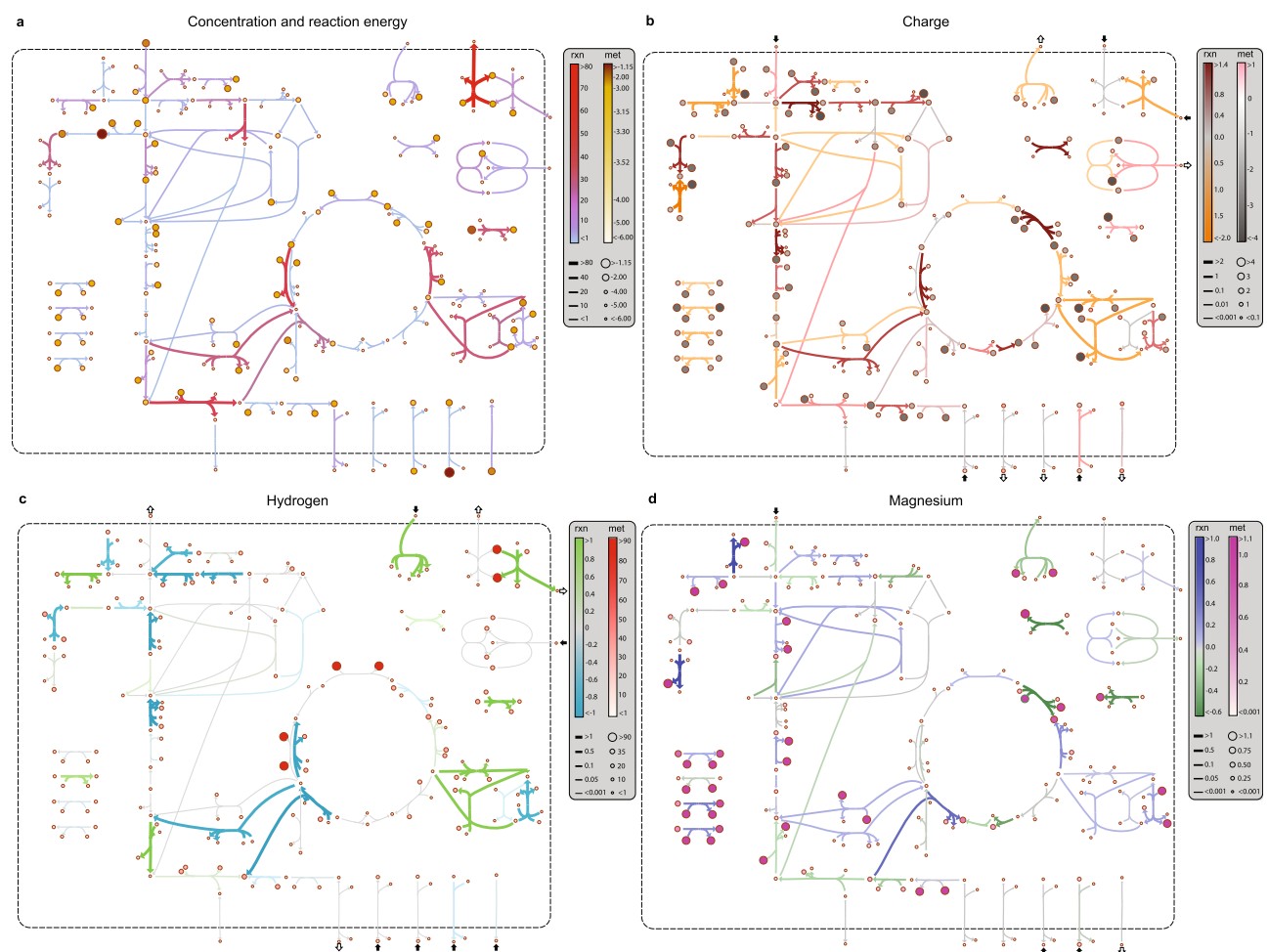

**Fig. 4 The ABC-based analysis elucidates the distinct ways, in which biochemical reactions perturb the physicochemical state of the cell.**
Computations are performed for exponential growth on glucose under physiological conditions ($pH_p = 7$, $pH_c = 7.5$, $[Mg]_c = 2$ mM, $I_c = 300$ mM).
**a** Concentration and energy map: reaction colorbar indicates transformed Gibbs energy of reactions in kJ/mol-rxn and metabolite colorbar indicates
$\log_{10}(C_i/C^o)$ with $C^o = 1$ M. **b** Charge map: reaction colorbar indicates intrinsic charge consumption in mol-e/mol-rxn and metabolite colorbar indicates
metabolite effective charge in mol-e/mol-met. **c** Hydrogen map: reaction colorbar indicates intrinsic hydrogen consumption in mol-H/mol-rxn and
metabolite colorbar indicates metabolite hydrogen content in mol-H/mol-met. **d** Magnesium map: reaction colorbar indicates intrinsic magnesium
consumption in mol-Mg/mol-rxn and metabolite colorbar indicates metabolite magnesium content in mol-Mg/mol-met. Metabolite and reaction colorbars
are denoted by "met" and "rxn", respectively. For **b**–**d**, arrows point in the direction, in which negative charge, hydrogen, and magnesium ions are
consumed. Reaction charge, hydrogen, and magnesium consumption for transport reactions, as indicated by the colorbars, do not reflect translocated
hydrogen ions. Small arrows next to each transport reaction show the direction of their respective net fluxes into and out of the cell, accounting for
translocated hydrogen ions and the accompanying metabolites. Enlarged and detailed maps are provided in Supplementary Figs. S8–S11. The definitions of
reaction charge, hydrogen, and magnesium consumption are given in "Abiotic constraints" section.

*E. coli* exhibits a stress response by taking up phosphate through
the *Pst* system to retain the phosphate influx, albeit at lower rates.
Note that, this is not the case for every membrane protein in *E.
coli*. Cytochrome oxidase in the ETC is an example, where the
transition from low-affinity (cytochrome-bo3) to high-affinity
(cytochrome-bd) system occurs most likely due to kinetic reasons.
Because these reactions are both highly energetic in their forward
direction ($\Delta_r G'^\circ \approx -90$ kJ/mol), their equilibria are never
reached for any biologically feasible concentration ranges of
ubiquinone, ubiquinol, and oxygen. Thus, the thermodynamic
constraints are unlikely to be responsible for the low activity of
cytochrome-bo3 in oxygen-limited environments.

**The ABCs shape transcriptional regulatory responses to
environmental stresses.** Biochemical reactions comprise trans-
formations among metabolites that can have multiple charge

states arising from hydrogen dissociation and metal-ion
binding[25]. These reactions liberate or bind hydrogen and metal
ions and, thus, interact with the cytoplasmic fluid to establish
intracellular pH and metal-ion homeostasis. The reactions that
have the highest rate of exchange with the cytoplasmic fluid
(highlighted in Fig. 4) represent the key players in maintaining
hydrogen and metal-ion concentrations. Thus, they are prime
targets for evolving transcriptional regulatory mechanisms in
response to environmental stresses.

Short-term transcriptional regulatory responses to acid and
osmotic stress are usually classified into phase I ($\lesssim 20$ min;
regulation is transient) and phase II (20–60 min; new steady state
is reached)[26]. Early stress responses by wild-type strains differ
from those exhibited by evolved strains that have adapted to high
osmolarities[27]. In our formulation, strain-specific characteristics
are captured by active pathways, dominant metabolite concen-
trations, and condition-specific parameters of the reduced

network. We constructed a charge map (Fig. 4b) using the characteristics of a wild-type *E. coli* strain to interpret short-term stress responses. Accordingly, we sought to establish associations between phase-I/II osmoregulatory responses (see "Known osmoregulation mechanisms" section) and reaction charge consumption provided by the ABC-based analysis (Fig. 4b).

From the RNA-sequencing data of Seo et al.[28], we identified 69 differentially expressed genes (>2-fold change 30 min after osmotic upshift) out of 124 genes associated with the reduced network. Among the primary-response genes that are differentially expressed, those associated with the ETC are all down-regulated and those with osmoprotectant transporters are upregulated. Although not explicitly included in the reduced network, potassium importers and sodium exporters are also activated. Moreover, several intracellular reactions are among the secondary regulatory targets. For example, glycolytic reactions (except GAPD) and glutamate decarboxylase (GLUDC) (not explicitly included in the reduced network) are activated, while glutamate synthase (GLUSy) and glutamate dehydrogenase (GLUDy) are suppressed.

These observations point to possible connections between reaction charge consumption and transcriptional regulatory responses to osmotic stress. These connections arise from cellular requirements to maintain electroneutrality, pH homeostasis, and a steady energy flow by coordinating glycolytic and ETC reactions. Qualitative connections have been drawn between expression data and ion homeostasis in the literature. For example, an inhibited respiration is regarded as an early response to an alkalinized cytoplasm due to a sudden efflux of hydrogen ions that accompanies potassium intake upon osmotic upshift[29]. This initial response is followed by a rapid accumulation of glutamate—a potassium counterion—through glutamate importers or its biosynthetic pathways[26,30]. Charge and pH neutrality are resumed in phase II upon flux regulations targeting the ETC, glycolysis, potassium and sodium transporters, glutamate transporters, glutamate biosynthesis, and glutamate decarboxylase. Here, the apparent inconsistency between the downregulated glutamate biosynthesis, observed from early stages[31], and elevated glutamate concentration is indicative of an active glutamate importer, such as glutamate/γ-aminobutyrate antiporter, in phase I.

To highlight the significant role that charge-balance and ion-binding constraints could play in transcriptional regulatory processes, we examined the regulation of glutamate in response to osmotic stress as a case study. We computed the intrinsic hydrogen and charge consumption (Methods: Eqs. (37) and (36); see "Abiotic constraints" section for the definition of intrinsic and extrinsic quantities) of glutamate-biosynthesis pathways (i.e., GLUDy, GLUSy, and GLUDC) and the glutamate/γ-aminobutyrate antiporter. We found that glutamate accumulation through these biosynthetic pathways does not affect the net cytoplasmic charge. It also lowers the hydrogen-ion concentration of an already alkalinized cytoplasm in phase I. In contrast, glutamate uptake through the glutamate/γ-aminobutyrate antiporter results in an inflow of negative charge, which can counterbalance the accumulated $K^+$ in the cytoplasm during phase I without affecting the pH, demonstrating the osmoregulatory role of glutamate antiporters (see "Glutamate role in osmoregulation" section for a more detailed discussion).

The interpretation of these complex processes can be simplified using graphical representations. The hydrogen (Fig. 4c) and magnesium (Fig. 4d) maps graphically visualize equilibrium shifts with respect to pH and pMg perturbations. Similarly to the charge map, these maps can help elucidate regulatory responses to pH- and pMg-related stress conditions. Consider the hydrogen map for example. Here, a change in $pH_c$ affects equilibrium constants,

Gibbs energy of reactions, and reaction rates according to La Chatelier's principle[25]. If a reaction produces hydrogen ions ($\Delta_r N_{H,j} < 0$) (Fig. 4c, blue arrows), then a positive perturbation in pH increases the equilibrium constant $K_j'$, which, in turn, shifts the reaction more in the forward direction, so that hydrogen ions are produced at a higher rate to counter the effect of the change. The magnesium map can be interpreted in a similar manner.

Thus, the ABCs shape transcriptional regulatory responses to environmental stresses. The combined effects of ion-binding and charge-balance constraints strongly restrict possible regulatory strategies to restore electroneutrality and pH homeostasis upon osmotic shock. In particular, the ABC-based analysis revealed why regulating the cytoplasmic charge through glutamate antiporters is more desirable than glutamate-biosynthesis pathways. This highlights the importance of the GAD operons in osmoregulation, the transcriptomic function of which has been recently detailed using large datasets and machine learning[32].

**Hydrogen-ion and charge imbalances underlie transcriptional regulatory responses to acid stress.** pH homeostasis is a fundamental cellular process. Bacteria can survive extreme acidic environments thanks to several sophisticated acid-resistance mechanisms, such as glucose-repressed-oxidative, glutamate-dependent, and arginine-dependent systems[33,34]. Acid stress causes increased hydrogen-ion entry into the cell, disrupting charge and hydrogen-ion balance. The cytoplasmic pH is perturbed as a result, affecting equilibrium constants, reaction energies, and reaction rates. This invokes transcriptional regulatory responses, similar to those observed in osmoregulation discussed above. Being directly involved in pH-homeostasis and reaction equilibria, reaction hydrogen consumptions are naturally expected to play a key role in acid-stress regulation.

To better understand how pH-dependent equilibrium shifts can affect transcriptional responses, we sought to identify connections between reaction hydrogen consumptions and gene expressions in phase-I/II RNA-sequencing data[34]. We identified 41 differentially expressed genes (>1.4-fold change within an hour after pH downshift) that are associated with the same reactions (except phosphate transporters) in the reduced network as those involved in osmoregulation. However, there are key differences in regulatory responses between the two conditions: Contrary to osmotic stress, NADH dehydrogenase is upregulated, while phosphate and osmoprotectant transporters are downregulated under acid stress. These gene-expression changes are consistent with the regulatory objective to neutralize the acidified cytoplasm by adjusting hydrogen-ion inflow and outflow rates.

For any given reaction, we found the intrinsic hydrogen consumption to be a reasonable indicator of its transcriptional regulatory state. Interestingly, among the 14 reactions with the highest hydrogen consumption or production ($|\Delta_r N_H| > 1$ mol-H/mol-rxn) in the hydrogen map (Fig. 4c), 11 are associated with the differentially expressed genes in phase I[35] and II[34]. To quantify the extent to which reaction hydrogen consumption in the reduced network can influence acid-stress response, we generated a connectivity map, identifying associations between key metabolic pathways and transcriptional regulators (Fig. 5). Glycolysis, glutamate metabolism, and the ETC are major regulatory targets in the reduced network. Their extrinsic hydrogen consumption in the base state is $\Delta_r N_H = -21.23$, 7.18, and 113.78 mmol-H/gDW/h, respectively. Three important points concerning this analysis are notable:

First, glycolysis is a major regulatory target. Overall, it is upregulated under acid and osmotic stress. Interestingly, it comprises reactions with high intrinsic hydrogen production (GAPD and PFK) and consumption (PYK). The same is true of

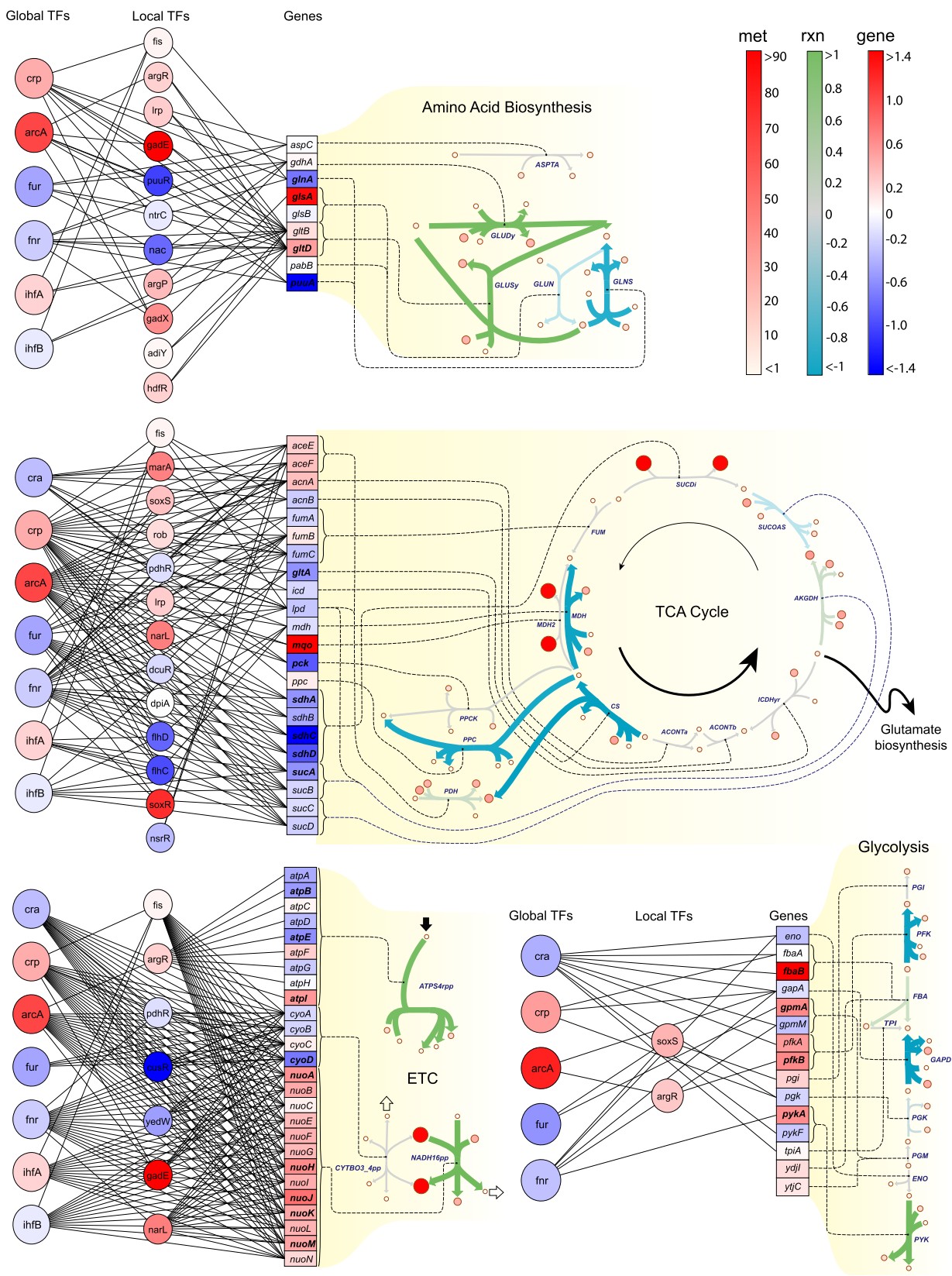

charge consumption and production. As a result, the net hydrogen consumption of the entire pathway ($\Delta_r N_H = -21.23$ mmol-H/gDW/h) is much smaller than that of the ETC ($\Delta_r N_H = 113.78$ mmol-H/gDW/h), even though the fluxes (obtained from FBA analysis) passing through these pathways are comparable. This feature allows the cell to reverse the flux direction through glycolysis in response to other simultaneous stresses (e.g., carbon-source change) without significant disruption to pH homeostasis (see "Glycolysis and gluconeogenesis in pH homeostasis" section).

Second, the glutamate-dependent system has been reported as the most effective acid-resistance mechanisms, complementing

**Fig. 5 Association of transcriptional regulatory responses with key metabolic pathways reveals factors influencing the _E. coli_ acid-stress response (pH$_{p,\epsilon}$ = 5.5).** Reaction colorbar "rxn" indicates hydrogen consumption in mol-H/mol-rxn, metabolite colorbar "met" indicates metabolite hydrogen content in mol-H/mol-met, and gene-expression colorbar "gene" indicates expression change log$_2$(TPM$_\epsilon$/TPM$_0$). The subscripts $\epsilon$ and 0 denote the acid-stress condition and base state. The base state corresponds to a stress-free growth in a neutral medium with pH$_{p,0}$ = 7, pH$_{c,0}$ = 7.5, [Mg]$_{c,0}$ = 2 mM, and $l_{c,0}$ = 300 mM. Genes with more than 1.4-fold expression change (shown in boldface) are considered differentially expressed. Expression data are taken from the RNA-sequencing analysis of Seo et al.[34] on a wild-type _E. coli_ strain. Reaction hydrogen consumption for the ETC reactions, as indicated by the colorbar, does not reflect translocated hydrogen ions. Small arrows next to each ETC reaction show the direction of net hydrogen-ion flux into (black) and out of (white) the cell.

primary responses[34]. Glutamate metabolism contains some of the highest hydrogen-consuming reactions in the reduced network—a possible explanation for their observed upregulation. They also carry high flux (especially GLUDy) whether or not the cell is under stress[29], so they can effectively attenuate acidification of the cytoplasm. The upper TCA cycle is downregulated, further promoting this mechanism by directing the flux of precursors from glycolysis and lower TCA towards glutamate biosynthesis.

As previously stated, these pathways are also involved in osmoregulation. However, unlike in pH regulation, GLUDy and GLUSy are both downregulated in response to osmotic stress. This raises a question as to why GLUDC is upregulated during osmoregulation when pH $\gtrsim$ 7.5. Here, contrasting transcriptional responses to osmotic and acid stresses in relation to hydrogen-ion balance might offer a plausible answer: Under osmotic stress, K$^+$ is the main glutamate counterion when the cytoplasm is alkalinized in phase I with GLUDC counterbalancing hydrogen-ion influx through sodium antiporters and osmoprotectant symporters in phase II. In contrast, under acid stress, hydrogen ion becomes one of the main glutamate counterions when the cytoplasm is acidic in phase I with GLUDC, GLUDy, and GLUSy offsetting hydrogen-ion leakage through the membrane in phase II.

Third, the ETC can contribute the most to hydrogen-ion balance in the cell. However, it is directly or indirectly coupled to many crucial cellular processes, such as the membrane potential, ion transport, energy balance, and redox balance. Therefore, regulation strategies that solely target the ETC would partially disrupt essential cellular functions and reduce the robust adaptability of bacteria to extreme and diverse environmental perturbations.

## Discussion

"Those [constraints] that operate at any given level are still valid at all more complex levels" — Francois Jacob[36]. Biological functions are subject to myriad constraints. The constraints on the function of abiotic and early biotic cells have been the subject of much discussion from Darwin's warm-pond hypothesis to the role of geothermal vents at the bottom of the ocean. Central to this discussion is the role of abiotic constraints. Given Francois Jacobs' quote, these constraints are fundamental and affect the function of ancestral and modern cells. The ABCs, thus, apply to functions across the biosphere.

In this study, we (i) formulated the ABCs, (ii) developed a computational platform, enabling their application to biological systems, (iii) showed that measured quantitative metabolomes satisfy these constraints, (iv) found that they furnish thermodynamic bottlenecks that mirror isotope-labeling experiments, (v) explained the reasons underlying the evolution of multiple phosphate transporters, and (vi) elucidated the constraints that shape transcriptional regulatory responses to external stresses. These results show the broad applications of the ABC-based analysis to understand biological functions and their origins. In fact, the two are inseparable.

Constraint-based approaches have significantly advanced in recent years, incorporating more detailed molecular descriptions of biological networks[7,37]. Despite their expanded predictive scope, these approaches still fall short in one key regard: Metabolite concentrations do not enter the formulation through governing equations or constraints grounded in first principles. To fill this gap, we introduced a constraint-based approach to characterize the concentration state of metabolic networks. This approach predicts feasible concentration and Gibbs-energy ranges resulting from the constraints biological networks are subject to.

We showed that the ABCs are consistent with metabolomic data across carbon sources. To investigate whether intracellular concentrations are associated with optimality principles, we studied two cellular objectives that have been previously considered[17], namely thermodynamic and enzyme-saturation efficiency. We found that biochemical reactions do not generally satisfy these objectives. Specifically, we identified several reactions from the TCA cycle and glycolysis that operate near their equilibrium during the exponential growth phase. Near-equilibrium reactions lower dissipative energy loss at higher protein cost. They also offer functional advantages, enabling rapid flux-direction reversal in energy-metabolism pathways to robustly achieve new steady states upon a wide variety of environmental perturbations. These observations underscore the complex and multifaceted nature of cellular metabolism, which cannot be fully described by single-objective or oversimplified optimality principles[5,38].

We studied the role of the ABCs in the evolution of phosphate transport systems. These concepts may be applied in a broader context to prebiotic reaction networks to gain insights into the origins and evolution of early metabolism[21,39]. The fundamental constraints we implemented in our analysis are particularly relevant to origins-of-life theories. Specifically, the membrane potential, transmembrane-ion gradients, charge balance, and thermodynamic laws are believed to have constrained the most ancient pathways for the formation of small organic molecules[21]. Several scenarios have been suggested for the origins of primitive life[40]. For all these theories to be plausible, one must establish a balance between the availability of an energy source that could have existed on the early earth (e.g., pH gradient in hydrothermal vents) and the energy required to operate the most energy demanding step (e.g., carbon-carbon bond formation) of the proposed chemistry for carbon metabolism (e.g., CO$_2$ reduction by H$_2$)[21]. These hypotheses can be rigorously formulated and examined within the quantitative framework we developed here to address some of the fundamental questions concerning the emergence of life from prebiotic chemistry.

Besides characterizing the quantitative metabolome, abiotic constraints shape regulatory responses to stress conditions. We analyzed osmotic- and acid-stress regulation to establish a relationship between transcriptional responses and reaction hydrogen and charge consumption. We argued that the net hydrogen and charge consumption of the electron transport chain, glutamate biosynthesis, and glutamate transporters can effectively counteract the hydrogen-ion and charge imbalances induced by osmotic

and acid stresses, providing an explanation for why they are subject to transcriptional regulation. Other reasons for why certain reactions are regulatory targets have been proposed. For example, reactions with a large thermodynamic driving force are believed to be under stricter transcriptional regulatory control than near-equilibrium ones since their rates are less sensitive to concentration perturbations[10]. Reaction rates that are sensitive to concentration variations can be effectively controlled through allosteric regulation. Hence, near-equilibrium reactions are expected to rely less on transcriptional regulations than highly exergonic ones.

An observed stress response is a resultant of overlapping allosteric and transcriptional regulatory processes, the organization and mechanism of which depend on: (i) regulation objective (e.g., restoring homeostasis after a vital constraint is violated), (ii) regulation constraints (e.g., charge and $H^+$ balance, osmotic balance, cell structural integrity, gene clusters and physical proximity, ATP and redox balance), and (iii) regulatory targets (biochemical reactions). We showed several ways, in which flux adjustments to biochemical reactions can modulate critical intracellular variables (net charge, hydrogen, and magnesium-ion concentrations, metabolite concentrations) to alleviate stress-induced imbalances, providing a more comprehensive description of stress-specific regulatory responses. The gene-expression activity of glycolytic reactions in response to acid stress is an example, where *pfkB* and *pykA* are differentially expressed. Although the corresponding reactions have the largest thermodynamic driving force, they also produce and consume the highest amount of $H^+$ among glycolytic reactions. Therefore, their transcriptional regulation is possibly more related to their role in pH homeostasis than their energetics.

The detailed formulation of the ABCs unraveled additional layers of attributes associated with biochemical reactions besides energetics and stoichiometry that are important to network functions. These constraints can help quantify perturbations in charge, pH, and pMg homeostasis induced by biochemical reactions, explaining the commonalities and differences in regulatory processes with similar objectives. Electroneutrality and pH homeostasis are examples of basic regulatory objectives that underlie transcriptional regulatory response to acid and osmotic stress, where we elucidated the associations between stress responses and charge-related constraints.

Taken together, the ABC-based analysis elucidates the role of physicochemical constraints in the operation of fundamental cellular processes, especially those involved in stress regulation. Thus, it lays the foundation for integrated models of flux, concentration, and macromolecular expression in the future that are capable of predicting functional states under any growth or stress conditions.

## Methods

**Reactants and species in biochemical reactions**. Most metabolites behave as weak acids in the cell, donating hydrogen ion to the intracellular fluid in multiple steps (ref. [25], Ch. 1). The resulting protonation states can, in turn, bind to several metal ions (e.g., $K^+$ and $Mg^{2+}$) in separate steps. These charge states, referred to as species, can coexist in the cell with varying distributions depending on the pH and ionic strength of the solution. Several species can be biologically active under physiological conditions, simultaneously participating in biochemical reactions. In general, all the protonation states and their magnesium-bound counterparts are active in the cell, especially those with phosphate groups (ref. [1], Sect. 9.4.2). We account for all the active charge states of a given metabolite, incorporating hydrogen-dissociation and magnesium-binding reactions (up to two $Mg^{2+}$)

$$H_k A \rightleftharpoons H + H_{k-1}A, \quad K_{0,k}, \quad k = 1 \cdots N, \quad (1)$$

$$MgH_k A \rightleftharpoons Mg + H_k A, \quad K_{1,k}, \quad k = 0 \cdots N, \quad (2)$$

$$Mg_2 H_k A \rightleftharpoons Mg + MgH_k A, \quad K_{2,k}, \quad k = 0 \cdots N \quad (3)$$

into our formulation. Here, $N$ is the number of hydrogen dissociation steps and A represents the minimum charge state of a metabolite in the cell with $K_{0,k}$, $K_{1,k}$, and $K_{2,k}$ the respective hydrogen-dissociation and magnesium-binding constants. These reactions tend to run faster than enzymatic reaction, always remaining close to their equilibrium (ref. [25], Ch. 1). This allows determining the distribution of species independently from the extension of biochemical reactions. Therefore, we simplify our analysis by lumping all the species together, representing them as a single reactant, with respect to which the abiotic constraints are to be expressed

$$\mathcal{A} := \{A, \cdots, H_N A, MgA, \cdots, MgH_N A, Mg_2 A, \cdots, Mg_2 H_N A\}. \quad (4)$$

Here, $\mathcal{A}$ represent a reactant in the cell, the concentration of which is given by

$$[\![A]\!] := [A] + \cdots + [H_N A] + [MgA] + \cdots + [MgH_N A] \\ + [Mg_2 A] + \cdots + [Mg_2 H_N A]. \quad (5)$$

For each species $A' \in \mathcal{A}$, the corresponding model fraction $\rho_{A'} := [A']/[\![A]\!]$ is calculated from hydrogen-dissociation and magnesium-binding constants, irrespective of the reaction $A'$ participates in (see "Composition of reactants" section). Accordingly, we define an effective charge

$$\bar{z}_{\mathcal{A}} := \sum_{A' \in \mathcal{A}} \rho_{A'} z_{A'} \quad (6)$$

for the reactant $\mathcal{A}$. Here, $z_{A'}$ denotes the charge of the species $A'$. This definition ensures the consistency of formulations with respect to reactants and species by requiring $\bar{z}_{\mathcal{A}}[\![A]\!]$ to furnish the total charge of all the species in $\mathcal{A}$. Although species always have an integral charge, reactants can generally carry fractional charges depending on the ionic strength, pH, and pMg of the solution because of their dependence on the mole fractions (Supplementary Fig. S1). The effective buffer intensity $\bar{\beta}_{\mathcal{A}}$, effective charge squared $\bar{\omega}_{\mathcal{A}}$, effective hydrogen content $\bar{N}_{H,\mathcal{A}}$, and effective magnesium content $\bar{N}_{Mg,\mathcal{A}}$ of $\mathcal{A}$ are defined similarly

$$\bar{\beta}_{\mathcal{A}} := \sum_{A' \in \mathcal{A}} \rho_{A'} \beta_{A'}, \quad (7)$$

$$\bar{\omega}_{\mathcal{A}} := \sum_{A' \in \mathcal{A}} \rho_{A'} z_{A'}^2, \quad (8)$$

$$\bar{N}_{H,\mathcal{A}} := \sum_{A' \in \mathcal{A}} \rho_{A'} N_{H,A'}, \quad (9)$$

$$\bar{N}_{Mg,\mathcal{A}} := \sum_{A' \in \mathcal{A}} \rho_{A'} N_{Mg,A'}. \quad (10)$$

In this article, the terms metabolite and reactant are interchangeably used. To distinguish between individual charge states and their aggregate representation in our formulation when necessary, we specifically refer to a metabolite and its charge states as reactant and species, respectively.

**Buffer capacity and buffer intensity**. The buffer capacity $\mathcal{B}$ is a measure of how much a strong base (or acid), such as NaOH, is needed to increase the pH of a solution. The buffer capacity of a mixture of reactants can be obtained from a superposition of the buffer capacities of individual reactants[41]. Therefore, we first derive an expression for the buffer capacity of a reactant $\mathcal{A}$. We consider all the charge states of $\mathcal{A}$ arising from hydrogen-dissociation and magnesium-binding reactions Eqs. (1)–(3) in our derivation. The charge balance for this system reads

$$z_{Na}[Na] + z_H[H] + z_{OH}[OH] + \sum_{A' \in \mathcal{A}} z_{A'}[A'] = 0. \quad (11)$$

Because NaOH is a strong base, it completely hydrolyzes, so that [Na] represents the concentration NaOH needed to neutralize $\mathcal{A}$. Note that, we did not explicitly state the charge that each ion carries in this equation for brevity—a notational convention we adopt throughout this document. We decompose [Na] into water $[Na]_w$ and reactant component $[Na]_{\mathcal{A}}$, which we refer to as the strong-base equivalent of $\mathcal{A}$, according to $[Na] = [Na]_w + [Na]_{\mathcal{A}}$, where

$$[Na]_w := \frac{K_w}{[H]} - [H], \quad (12)$$

$$[Na]_{\mathcal{A}} := -[\![A]\!] \sum_{A' \in \mathcal{A}} \rho_{A'} z_{A'} \quad (13)$$

with $K_w$ the water dissociation constant. Accordingly, the buffer capacity of water and reactant $\mathcal{A}$ are defined[41]

$$\mathcal{B}_w := \frac{d[Na]_w}{dpH} = \ln(10) \left( \frac{K_w}{[H]} + [H] \right), \quad (14)$$

$$\mathcal{B}_{\mathcal{A}} := \frac{\partial[Na]_{\mathcal{A}}}{\partial pH} = -[\![A]\!] \sum_{A' \in \mathcal{A}} \frac{\partial \rho_{A'}}{\partial pH} z_{A'}. \quad (15)$$

The effective buffer intensity of $\mathcal{A}$ is simply defined as $\bar{\beta}_{\mathcal{A}} := \mathcal{B}_{\mathcal{A}}/[\![A]\!]$. The buffer capacity can be approximated by linearizing $\mathcal{B}_{\mathcal{A}}$ around a pH of interest. It is correspondingly defined as the amont of a strong base one should add to a solution to increase its pH by one. Supplementary Fig. S2 illustrates how the strong-base equivalent and buffer intensity of $\mathcal{A}$ vary with pH. The total buffer capacity of a mixture of reactants is determined from those of the individual reactants

$$\mathcal{B}_{\mathrm{T}} = \mathcal{B}_{\mathrm{w}} + \sum_{i \in \mathcal{I}} \mathcal{B}_i, \qquad (16)$$

where $\mathcal{I}$ is the index set of reactants in the solution.

**Osmotic pressure and activity models.** Osmosis is a prevalent phenomenon in electrolyte systems, where there are ion-concentration differentials. Consider the osmotic coefficient of a multicomponent aqueous solution[42]

$$\phi := \frac{\Pi}{C_{\mathrm{t}} R T} = -\frac{\vartheta_{\mathrm{w}}}{C_{\mathrm{t}}} \ln a_{\mathrm{w}}, \qquad (17)$$

where $C_{\mathrm{t}}$ is the total molar concentration of solutes, $R$ gas universal constant, and $T$ temperature. Here, $\vartheta_{\mathrm{w}}$, $a_{\mathrm{w}}$, and $x_{\mathrm{w}}$ denote molar density, activity, and mole fraction of water. The osmotic pressure $\Pi$ measures the pressure of the solution relative to that of pure water at the same temperature. We use Pitzer's model for the water activity[43]

$$\ln a_{\mathrm{w}} = -\left(\frac{C_{\mathrm{t}}}{\vartheta_{\mathrm{w}}}\right)\left(1 + \frac{2 I_{\mathrm{m}}}{m_{\mathrm{t}}} f^{\phi}(I_{\mathrm{m}})\right) + \mathcal{O}\left(\frac{m_i^2}{m_{\mathrm{t}}}\right), \qquad (18)$$

where

$$f^{\phi}(I_{\mathrm{m}}) := -\frac{A^{\phi} \sqrt{I_{\mathrm{m}}}}{1 + B_{\mathrm{PZ}}^{\phi} \sqrt{I_{\mathrm{m}}}} \qquad (19)$$

with $A^{\phi} = 0.391475$ kg$^{1/2}$ mol$^{-1/2}$, $B_{\mathrm{PZ}}^{\phi} = 1.2$ kg$^{1/2}$ mol$^{-1/2}$, $m_{\mathrm{t}}$ total molal concentration of solutes, and $I_{\mathrm{m}}$ the molal ionic strength[43]. We neglect the second and higher order terms in $m_i$, corresponding to ion-ion interactions, in Eq. (18). Since the parameters required to estimate these interactions are not generally known for biological systems, they are usually neglected[25]. Satisfactory results have been reported in equilibrium studies of biochemical reactions using activity models based on this approximation in concentration ranges of physiological relevance, justifying this assumption[25]. Molal concentrations can be converted to molar concentrations according to $C_i = \hat{\rho}_{\mathrm{w}} m_i$, where $C_i$ and $m_i$ are molar and molal concentrations of solute $i$. Moreover, $\hat{\rho}_{\mathrm{w}} = \rho_{\mathrm{w}} - \sum_{i \in \mathcal{I}_s} C_i \mathrm{MW}_i$, where $\hat{\rho}_{\mathrm{w}}$ and $\rho_{\mathrm{w}}$ are the reduced density and density of solution with $\mathcal{I}_s$ and $\mathrm{MW}_i$ the index set of solutes in the solution and molecular weight of solute $i$. In general, these densities are functions of temperature, pressure, and solute concentrations. However, in biological systems, they are commonly approximated by the density of water since solute concentrations in the cell are negligible compared to water[25]. We adopt this approximation to convert between molar and molal concentrations through a multiplicative constant. Note that, if a molar ionic strength $I$ is passed to $f^{\phi}$, the constants in Eq. (19) must be adjusted according to $A^{\phi} \mapsto A^{\phi}/\sqrt{\hat{\rho}_{\mathrm{w}}}$ and $B_{\mathrm{PZ}}^{\phi} \mapsto B_{\mathrm{PZ}}^{\phi}/\sqrt{\hat{\rho}_{\mathrm{w}}}$. Given the relationship $I_{\mathrm{m}} = I/\hat{\rho}_{\mathrm{w}}$, these ensure the consistency of ionic-strength and parameter units. Substituting Eq. (18) in Eq. (17), we derive an expression for the osmotic coefficient

$$\phi = 1 + \frac{2 I}{C_{\mathrm{t}}} f^{\phi}(I), \qquad (20)$$

where molality-to-molarity conversion has been applied. We use this expression to estimate the osmotic pressure of the cytoplasmic and periplasmic fluids in our model

$$\frac{\Pi_{\mathrm{c}}}{R T} = C_{\mathrm{t,c}} + 2 I_{\mathrm{c}} f^{\phi}(I_{\mathrm{c}}), \qquad (21)$$

$$\frac{\Pi_{\mathrm{p}}}{R T} = C_{\mathrm{t,p}} + 2 I_{\mathrm{p}} f^{\phi}(I_{\mathrm{p}}), \qquad (22)$$

where the subscripts c and p denote the quantities associated with the cytoplasm and periplasm.

The activity coefficients of solutes are also needed for equilibrium computations that will be discussed in subsequent sections. The activity coefficient of charge solutes in ionic solutions can be generally expressed as

$$\ln \gamma_i^{(\mathrm{m})} = -z_i^2 f^{\gamma}(I_{\mathrm{m}}) + \mathcal{O}(m_i), \qquad (23)$$

where $\gamma_i^{(\mathrm{m})}$ is the molality-based activity coefficient. Two widely accepted activity models are the extended Debye–Hückel and Pitzer, respectively given by[25,43]

$$f^{\gamma}(I_{\mathrm{m}}) := \frac{3 A^{\phi} \sqrt{I_{\mathrm{m}}}}{1 + B_{\mathrm{DH}}^{\phi} \sqrt{I_{\mathrm{m}}}}, \qquad (24)$$

$$f^{\gamma}(I_{\mathrm{m}}) := A^{\phi}\left[\frac{\sqrt{I_{\mathrm{m}}}}{1 + B_{\mathrm{PZ}}^{\phi} \sqrt{I_{\mathrm{m}}}} + \frac{2}{B_{\mathrm{PZ}}^{\phi}} \ln\left(1 + B_{\mathrm{PZ}}^{\phi} \sqrt{I_{\mathrm{m}}}\right)\right], \qquad (25)$$

where $B_{\mathrm{DH}}^{\phi} = 1.6$ kg$^{1/2}$ mol$^{-1/2}$ [44]. As with the osmotic coefficient, the parameter transformations $A^{\phi} \mapsto A^{\phi}/\sqrt{\hat{\rho}_{\mathrm{w}}}$, $B_{\mathrm{PZ}}^{\phi} \mapsto B_{\mathrm{PZ}}^{\phi}/\sqrt{\hat{\rho}_{\mathrm{w}}}$, and $B_{\mathrm{DH}}^{\phi} \mapsto B_{\mathrm{DH}}^{\phi}/\sqrt{\hat{\rho}_{\mathrm{w}}}$ apply if $I$ is passed to $f^{\gamma}$ instead of $I_{\mathrm{m}}$. Our formulations in the forthcoming sections are with respect to molarity-based activity coefficients $\gamma_i$. These are related to molality-based activity coefficients by $a_i := \gamma_i C_i/C^{\circ} := \gamma_i^{(\mathrm{m})} m_i/m^{\circ}$, where $C^{\circ} = 1$ mol/L-w

and $m^{\circ} = 1$ mol/kg-w are the standard concentrations. As will be discussed, activity coefficients are needed to calculate the formation energies of species at a given $I$ from standard formation energies, which we approximate using group-contribution methods[45]. These techniques estimate group-contribution parameters using the extended Debye–Hückel model by minimizing the difference between the predicted and measured equilibrium constants of biochemical reactions. Since equilibrium constants are highly sensitive to these parameters, we adhere to the extended Debye–Hückel model in our formulation.

**Abiotic constraints.** Metabolism is a hallmark of living systems. It is orchestrated through a delicate balance between metabolite concentrations, biochemical-reaction fluxes, and macromolecular levels in the cell. Metabolic networks are tightly intertwined with complex, overlapping regulatory and signaling networks, enabling the cell to sustain homeostasis or adapt to new environments by transitioning between homeostatic states[14]. Quantitative description of dynamical systems of such complexity using mechanistic kinetic models is often restricted to small networks[46] due to computational challenges, limited kinetic data, and incomplete knowledge about the mechanisms of enzymatic reactions[47].

We introduce a constrained-based formalism to characterize the metabolome. Rather than specifying a unique time-dependent concentration state, we seek a set of concentration states that respect all the constraints biological networks are subject to, allowing for a unified treatment of steady-state and oscillatory dynamics. Fundamental constraints, such as the law of mass action, charge balance, and osmotic balance, imposed by the laws of thermodynamics, electroneutrality, and osmotic pressure, are obeyed by all electrochemical systems. The law of mass action, which we refer to as thermodynamic constraints in the main text, arises from the dissipative structure of chemical-reaction systems and the second law of thermodynamics[48,49], while the charge and osmotic balances are essential for cell-volume and ion-transport stability[50]. Evolutionary constraints, such as fixed ionic strength and buffer capacity, have emerged from the evolutionary adaptation of biological networks. The ionic strength plays a crucial role in several cellular processes, such as osmoregulation, enzyme activity, and protein structure[51], while the buffer capacity is associated with pH homeostasis[52]. These condition-specific parameters are controlled by regulatory networks to maintain a stable metabolism under various charge-related stress conditions.

The complexities of regulatory networks impede the application of mechanistic dynamic models, even for those controlling the simplest cellular processes[14]. Therefore, we adopt a top-down approach, where the phenotypes arising from important regulatory processes (e.g., osmoregulation and ion homeostasis) are implicitly accounted for by incorporating them as additional constraints (the evolutionary constraints) into our model. The foregoing fundamental and evolutionary constraints are mathematically expressed as

$$\sum_{i \in \mathcal{I}_{\mathrm{c}}} \bar{z}_i C_i + \sum_{i \in \mathcal{J}_{\mathrm{c}}} \bar{z}_i C_i = \zeta_{\mathrm{c}}, \quad \text{(Charge balance)} \qquad (26)$$

$$\sum_{i \in \mathcal{I}_{\mathrm{c}}} \bar{\omega}_i C_i + \sum_{i \in \mathcal{J}_{\mathrm{c}}} \bar{\omega}_i C_i = 2 I_{\mathrm{c}}, \quad \text{(Ionic strength)} \qquad (27)$$

$$\sum_{i \in \mathcal{I}_{\mathrm{c}}} \bar{\beta}_i C_i + \sum_{i \in \mathcal{J}_{\mathrm{c}}} \bar{\beta}_i C_i = \mathcal{B}_{\mathrm{c}} - \mathcal{B}_{\mathrm{w}}, \quad \text{(Buffer capacity)} \qquad (28)$$

$$\sum_{i \in \mathcal{I}_{\mathrm{c}}} C_i + \sum_{i \in \mathcal{J}_{\mathrm{c}}} C_i = C_{\mathrm{t,c}}, \quad \text{(Bounded cytoplasm)} \qquad (29)$$

$$C_{\mathrm{t,c}} - C_{\mathrm{t,p}} + 2\left[I_{\mathrm{c}} f^{\phi}(I_{\mathrm{c}}) - I_{\mathrm{p}} f^{\phi}(I_{\mathrm{p}})\right] = \frac{\Delta \Pi}{R T}, \quad \text{(Osmotic balance)} \qquad (30)$$

$$\Delta_r G_j^{\prime \circ} + R T \ln \Gamma_j \leq 0, \quad j = 1 \cdots m, \quad \text{(Thermodynamics)} \qquad (31)$$

$$C_i \sim K_{m,i,j}, \quad i \in \mathcal{I}_{\mathrm{c}}, \quad j = 1 \cdots m \quad \text{(Enzyme saturation)} \qquad (32)$$

with $\mathbf{C}$ the vector of cytoplasmic reactant concentrations ($C_{\mathcal{A}}$ and [[A]] are interchangeably used to denote the concentration of $\mathcal{A}$. For a reactant with only one charge state, [A] and [[A]] are indistinguishable), $C_{\mathrm{t,c}}$ total cytoplasmic concentration, $C_{\mathrm{t,p}}$ total periplasmic concentration, $\zeta$ net cytoplasmic charge, $I_{\mathrm{c}}$ cytoplasmic ionic strength, $I_{\mathrm{p}}$ periplasmic ionic strength, $\mathcal{B}_{\mathrm{c}}$ total cytoplasmic buffer capacity, $\mathcal{B}_{\mathrm{w}}$ buffer capacity of water, $\Delta \Pi := \Pi_{\mathrm{c}} - \Pi_{\mathrm{p}}$ osmotic-pressure differential, $f^{\phi}$ Pitzer's function for osmotic coefficient [Eq. (5)][43], $\Delta_{\mathbf{r}} \mathbf{G}^{\prime \circ}$ vector of standard transformed Gibbs energy of reaction, $\Gamma$ reaction quotient, $K_{\mathrm{m}}$ Michaelis constant, $j$ reaction index, $i$ reactant index, and $m$ number of reactions. Moreover, $\mathcal{I}_{\mathrm{c}}$ and $\mathcal{J}_{\mathrm{c}}$ are the index sets of cytoplasmic reactants and transmembrane ions (K$^+$, Na$^+$, Cl$^-$, Mg$^{2+}$, OH$^-$, and H$^+$)[53] affecting the membrane potential $\Delta \psi$. Similarly, the index sets of periplasmic reactants and transmembrane ions are denoted $\mathcal{I}_{\mathrm{p}}$ and $\mathcal{J}_{\mathrm{p}}$. We applied these constraints to characterize the CSS of a reduced metabolic network of *E. coli* (Supplementary Fig. S3).

Equations (26)–(31) represent charge balance, fixed ionic strength, fixed buffer capacity, bounded cytoplasmic metabolome, osmotic balance, and the law of mass action, respectively. Equation (32) corresponds to the enzyme-saturation constraint (see "Enzyme kinetics" section). In general, we did not impose it as a hard constraint because it does not hold for every reaction in the network (e.g., amino-acid degradation reactions[13]). However, it is used to evaluate the saturation level of

enzyme and how well it describes the functional state of the cell by comparing metabolomic date with $K_m^{max}$ values. Nonetheless, in Fig. 1b, feasible concentration ranges when enzyme saturation is imposed as a hard constraint, in which case dark gray bars are infeasible, and when it is not, in which case dark gray bars are feasible, are both determined and compared.

The standard transformed Gibbs energy $\Delta_r G_j^{'\circ}$ of reaction $j$ is obtained from the standard transformed Gibbs energy of formation $\Delta_f G_i^{'\circ}$ of reactants $i \in \mathcal{I}_c \cup \mathcal{I}_p$ according to their coefficients in the stoichiometric matrix $\mathbf{S}$. The intrinsic reaction hydrogen consumption $\Delta_r N_{H,j}$, magnesium consumption $\Delta_r N_{Mg,j}$, and charge consumption $\Delta_r Z_j$ are similarly defined. Reactions with $\Delta_r Z_j > 0$ and $\Delta_r Z_j < 0$ consume and produce negative charge in the cytoplasm by convention, respectively. In the main text, we generally refer to $\Delta_r Z_j$ as the reaction charge consumption, irrespective of its sign. When necessary, we specifically refer to it as charge consumption or charge production to emphasize the sign of $\Delta_r Z_j$. The same conventions apply to $\Delta_r N_{H,j}$ and $\Delta_r N_{Mg,j}$. We use the same notation to denote extrinsic reaction hydrogen, magnesium, and charge consumption. These are defined as the product of their intrinsic counterparts and the respective flux. We clarify whichever is relevant in the context by explicitly stating their units (intrinsic and extrinsic consumptions are measured in mol/mol-rxn and mmol/gDW/h, recetively).

Since biochemical reactions involve transformations between reactants with multiple charge states that exchange ions with the cytoplasmic and periplasmic fluids, they generally do not balance charge, hydrogen, and magnesium ions. Indeed, the reaction charge, hydrogen, and magnesium consumption discussed above are defined to quantify this property. In the following discussion, we make these definitions more precise. Starting with charge consumption, we introduce three components for $\Delta_r Z_j$

$$\Delta_r Z_j^{(S)} := \begin{cases} 0 & \text{Exporters,} \\ -\sum_{i \in \mathcal{I}_c \cup \mathcal{I}_p} S_{ij} \bar{z}_i & \text{Otherwise,} \end{cases} \tag{33}$$

$$\Delta_r Z_j^{(T)} := \begin{cases} \sum_{i \in \mathcal{I}_c} S_{ij} \bar{z}_i & \text{Exporters,} \\ \sum_{i \in \mathcal{I}_p} S_{ij} \bar{z}_i & \text{Importers,} \\ 0 & \text{Otherwise,} \end{cases} \tag{34}$$

$$\Delta_r Z_j^{(H)} := \tau_{H,j} \tag{35}$$

with $\tau_{H,j}$ the number of translocated hydrogen ions in reaction $j$, where $\tau_{H,j} = 0$ for intracellular reactions, $\tau_{H,j} > 0$ if hydrogen ion is translocated from the periplasm to cytoplasm, and $\tau_{H,j} < 0$ if hydrogen ion is translocated from the cytoplasm to periplasm. Exporters are thermodynamically spontaneous in the direction, where reactants are transferred from the cytoplasm to periplasm, and importers are thermodynamically spontaneous in the opposite direction. Note that, $\Delta_r Z_j^{(S)}$ reflects the amount of charge exchanges between reactants and water due to hydrogen dissociation and magnesium binding. For intracellular reactions, charge exchanges occur within the cytoplasmic fluid. However, ambiguity may arise for transport reactions as to whether reactants exchange charge with the cytoplasmic or periplasmic fluids. To avoid this ambiguity, we assume that these exchanges occur on the product side of transport reactions (Supplementary Fig. S4). The definition given by Eq. (34) is in accordance with this assumption. The reaction charge consumption is defined

$$\Delta_r Z_j := \Delta_r Z_j^{(S)} + \Delta_r Z_j^{(T)} + \Delta_r Z_j^{(H)}. \tag{36}$$

We term $\Delta_r Z_j^{(S)}$, $\Delta_r Z_j^{(T)}$, and $\Delta_r Z_j^{(H)}$ the stoichiometry, reactant-transport, and hydrogen-transport components. According to this definition, $\Delta_r Z_j$ does not affect the overall charge balance of the cytoplasmic fluid for intracellular reactions. However, for transport reactions, it measures the net negative charge flowing into or out of the cell.

Reaction hydrogen consumption is an important quantity, elucidating the role of biochemical reactions in pH homeostasis. To properly define it, a key difference between $\Delta_r Z_j$ and $\Delta_r N_{H,j}$ must be accounted for: While the goal of $\Delta_r Z_j$ is to capture how biochemical reactions affect reactant-cytoplasmic fluid charge transfer and the overall charge balance of the cell, $\Delta_r N_{H,j}$ is defined to measure the contribution of a reaction to pH$_c$. This is based on the assumption that maintaining a near-neutral pH$_c$ is a more essential constraint for the operation of biological networks than the overall hydrogen balance of the cell. Accordingly, reaction hydrogen consumption is defined by two components

$$\Delta_r N_{H,j} := \Delta_r N_{H,j}^{(S)} + \Delta_r N_{H,j}^{(H)}, \tag{37}$$

where

$$\Delta_r N_{H,j}^{(S)} := \begin{cases} 0 & \text{Exporters,} \\ \sum_{i \in \mathcal{I}_c \cup \mathcal{I}_p} S_{ij} \bar{N}_{H,j} + 2 S_{w,j} & \text{Otherwise.} \end{cases} \tag{38}$$

$$\Delta_r N_{H,j}^{(H)} := -\tau_{H,j} \tag{39}$$

with $S_{w,j}$ the stoichiometric coefficient of water in reaction $j$ (see "Standard transformed Gibbs energy of reaction" section). As with charge consumption, hydrogen-ion exchanges between reactants and water is assumed to occur on the product side of transport reactions in Eq. (38). Reaction magnesium consumption is defined in a similar manner

$$\Delta_r N_{Mg,j} := \Delta_r N_{Mg,j}^{(S)}, \tag{40}$$

where

$$\Delta_r N_{Mg,j}^{(S)} := \begin{cases} 0 & \text{Exporters,} \\ \sum_{i \in \mathcal{I}_c \cup \mathcal{I}_p} S_{ij} \bar{N}_{Mg,j} & \text{Otherwise.} \end{cases} \tag{41}$$

In Eqs. (26)–(32), the concentration of cytoplasmic reactants, excluding the transmembrane ions, are the unknowns. The transmembrane-ion concentrations, $I_c$, and $\mathcal{B}_c$ are condition- and strain-specific, the values of which are specified at the outset (Supplementary Table S2). All other quantities are the given parameters of the model (Supplementary Table S2). Periplasmic concentrations are assumed to be identical to those of the growth medium with a specified composition (M9 minimal medium[54], detailed in Supplementary Table S1).

We define the CSS as a subset of the concentration space, where all the biophysical constraint are satisfied:

$$\mathcal{C} := \left\{ \mathbf{C}_c \in \mathbb{R}_+^n : \text{Eqs. (26)−(31) hold} \right\} \tag{42}$$

with $n := |\mathcal{I}_c|$, $n' := |\mathcal{I}_p|$, and $\mathbf{C}_c$ a subvector of $\mathbf{C}$ corresponding to $\mathcal{I}_c$. Geometrically, the CSS can be represented as the intersection of the affine subspace defined by the equality constraints Eqs. (26)–(30) and the thermodynamically concentration solution space corresponding to Eq. (31) (Fig. 1b). We characterize the CSS, which can generally be nonconvex and disconnected, using global optimization and sampling techniques. The first furnishes global bounds on metabolite concentrations and reaction energies, regardless of whether the CSS is connect or not. The second provides expectations of concentrations $\mathbb{E}(C_i)$ and reaction energies $\mathbb{E}(\Delta_r G_j')$, standard deviations of concentrations $\sigma(C_i)$ and reaction energies $\sigma(\Delta_r G_j')$, and violation probabilities of thermodynamic constraints $P_j$. Here, we first identify an interior point $q$ by solving a parametric optimization problem (Eq. (85)) that maximally distances $q$ from the thermodynamic constraints, simultaneously avoiding arbitrarily small concentrations, which are biologically irrelevant. We found these solutions to be well-correlated with metabolomic data reported in the literature. We then explore a neighborhood of $q$ by generating curves along random directions inside the CSS to sample the space and determine the thermodynamic constraints that are violated more frequently (see "Characterization of Concentration Solution Space" section).

We reformulate Eqs. (26)–(31) with respect to mole fractions to arrive at a compact dimensionless form

$$\begin{cases} \mathbf{A}\mathbf{x} = \mathbf{f}^{eq}, \\ C_{t,c} - C_{t,p}^* + 2\left[ I_c^* f^\phi(I_c^*) - I_p^* f^\phi(I_p^*) \right] = 1, \\ \mathbf{S}_c^T \ln \mathbf{x} \leq \mathbf{f}^{ineq}, \\ \mathbf{x} \geq \mathbf{0} \end{cases} \tag{43}$$

with

$$\mathbf{A} := \begin{bmatrix} \bar{\mathbf{z}}^T \\ \bar{\boldsymbol{\omega}}^T \\ \bar{\boldsymbol{\beta}}^T \\ \mathbf{1}_{1 \times n} \end{bmatrix}, \quad \mathbf{f}^{eq} := \begin{bmatrix} -(\sum_{i \in \mathcal{J}_c} \bar{z}_i C_i^*)/C_{t,c}^* \\ (2I_c^* - \sum_{i \in \mathcal{J}_c} \bar{\omega}_i C_i^*)/C_{t,c}^* \\ (\mathcal{B}_c^* - \mathcal{B}_w^* - \sum_{i \in \mathcal{J}_c} \bar{\beta}_i C_i^*)/C_{t,c}^* \\ 1 - (\sum_{i \in \mathcal{J}_c} C_i^*)/C_{t,c}^* \end{bmatrix}, \tag{44}$$

and

$$\mathbf{f}^{ineq} := -\begin{bmatrix} \frac{\Delta_r G_1^{'\circ}}{RT} + v_1 \ln\left(\frac{C_s}{C^\circ}\right) + v_1^{(c)} \ln C_{t,c}^* + \sum_{i \in \mathcal{I}_p} S_{i1} \ln C_i^{'*} \\ \vdots \\ \frac{\Delta_r G_m^{'\circ}}{RT} + v_m \ln\left(\frac{C_s}{C^\circ}\right) + v_m^{(c)} \ln C_{t,c}^* + \sum_{i \in \mathcal{I}_p} S_{im} \ln C_i^{'*} \end{bmatrix}, \tag{45}$$

where all the concentrations, ionic strengths, and buffer capacities are scaled with $C_s := \Delta\Pi/RT$. Here, $\mathbf{x} := \mathbf{C}_c/C_{t,c}$ is the vector of cytoplasmic mole fractions, the superscript * denotes scaled parameters or scaled variable, $\mathbf{C}'$ is the vector of periplasmic reactant concentrations, $\mathbf{S}_c$ is a submatrix of $\mathbf{S}$ containing all the rows corresponding to $\mathcal{I}_c$, $v_j := \sum_{i \in \mathcal{I}_c \cup \mathcal{I}_p} S_{ij}$, $v_j^{(c)} := \sum_{i \in \mathcal{I}_c} S_{ij}$, $\mathbf{x} \in \mathbb{R}^n$, $\mathbf{S} \in \mathbb{R}^{(n+n') \times m}$, $\mathbf{A} \in \mathbb{R}^{\ell \times n}$, $\mathbf{f}^{eq} \in \mathbb{R}^\ell$, $\mathbf{f}^{ineq} \in \mathbb{R}^m$, $\mathbf{v} \in \mathbb{R}^m$, $\mathbf{v}^{(c)} \in \mathbb{R}^m$ and $\mathbf{S}_c \in \mathbb{R}^{n \times m}$ with $\ell$ the number of linear equalities (i.e., rows of $\mathbf{A}$) in Eq. (43). Note that, because $I^*$ is passed to $f^\phi$ instead of $I_m$ in Eq. (43), the constants of Eq. (19) must be adjusted according to $A^\phi \mapsto A^\phi \sqrt{C_s/\bar{\rho}_w}$ and $B_{PZ}^\phi \mapsto B_{PZ}^\phi \sqrt{C_s/\bar{\rho}_w}$. In Eq. (43), $\mathbf{x}$ and $C_{t,c}^*$ are unknown variables; all the other parameters are known and specified at the outset. The CSS in the mole-fraction space is defined

$$\mathcal{X} := \left\{ \mathbf{x} \in [0,1]^n : \text{Eq. (43) holds} \right\}. \tag{46}$$

The CSS spans many orders of magnitude in the concentration and mole-fraction spaces. It has a highly irregular shape in these spaces, extending several orders of magnitude wider in some directions than others. Therefore, geometric characterization of the CSS (e.g., sampling, identifying the global minimum of a function over the CSS, or constructing bounding box) poses major computational challenges. Accumulation of large rounding errors is one of the main computational issues that arise from systems involving variables with contrasting magnitudes. To overcome these challenges, we use the logarithmic map $\Xi := \mathbf{x} \mapsto \mathbf{y} = \ln \mathbf{x}$, reformulating Eqs. (43) and (46) into the forms

$$\begin{cases} \mathbf{A}\exp\mathbf{y} = \mathbf{f}^{eq}, \\ C_{t,c}^* - C_{t,p}^* + 2\left[I_c^*f^\phi(I_c^*) - I_p^*f^\phi(I_p^*)\right] = 1, \\ \mathbf{S}_c^T\mathbf{y} \le \mathbf{f}^{ineq}, \\ \mathbf{y} \le \mathbf{0}, \end{cases} \quad (47)$$

$$\mathcal{Y} := \left\{\mathbf{y} \in \mathbb{R}_-^n : \text{Eq. (47) holds}\right\}, \quad (48)$$

in which it is more computationally tractable to characterize the CSS. We refer to the mole-fraction and logarithmic mole-fraction spaces as the $X$ and $Y$ space, respectively.

**Enzyme kinetics.** The reversible Michaelis–Menten mechanism[55]

$$\nu_1^- S_1 + \cdots + \nu_{n_s}^- S_{n_s} + E \rightleftharpoons S_1 \cdots S_{n_s} E \rightleftharpoons EP_1 \cdots P_{n_p} \rightleftharpoons E + \nu_1^+ P_1 + \cdots + \nu_{n_p}^+ P_{n_p} \quad (49)$$

is one of the simplest mechanisms proposed to study the kinetics of biochemical reactions. It leads to the separable rate law[55]

$$\nu = [E]k_{cat}^+ \kappa \varsigma, \quad (50)$$

where

$$\kappa := \frac{\prod_{i=1}^{n_s}([S_i]/K_{s,i})^{\nu_i^-}}{1 + \prod_{i=1}^{n_s}([S_i]/K_{s,i})^{\nu_i^-} + \prod_{i=1}^{n_p}([P_i]/K_{p,i})^{\nu_i^+}}, \quad (51)$$

and

$$\varsigma := 1 - \exp(\Delta_r G'/RT) \quad (52)$$

are the enzyme-saturation and thermodynamic efficiencies. Here, $K_{s,i}$ and $K_{p,i}$ are the Michaelis constants of the substrates and products with $\kappa$ the enzyme-saturation efficiency, $\varsigma$ thermodynamic efficiency, $[E]$ total enzyme concentration, $k_{cat}^+$ the turnover number, $n_s$ the number of substrates, and $n_p$ the number of products. Note that the specific form of Eqs. (50)–(52) depends on the rate law describing the individual steps of the underlying mechanism, and the assumption that enzyme complexes always remain at a steady state. It is clear from Eq. (50) that to draw maximum flux at minimum protein cost, both efficiencies must be maximized. However, the maximum saturation efficiency (i.e., $\kappa \to 1$ achieved when $[S_i] \gg K_{s,i}$ and $[P_i] \gg K_{p,i}$) cannot be realized by every reaction in the network because an unbounded increase in all substrate and product concentrations may conflict with other constraints imposed on the network, such as osmotic balance and molecular crowding. Therefore, a more relaxed efficiency criterion is usually used, where a reaction is considered efficient when it is at least half-saturated in one direction ($[S_i] \ge K_{s,i}$ in the forward or $[P_i] \ge K_{p,i}$ in the backward direction)[13,17]. Reversible reactions are required to be half-saturated or more in both directions to be considered efficient. For reactant $i$, we define the maximum Michaelis constant

$$K_{m,i}^{max} := \max_{j \in \{1 \cdots m\}} K_{m,i,j}, \quad (53)$$

where $K_{m,i,j}$ stands for $K_{s,i}$ and $K_{p,i}$ in Eq. (51) that are associated with reaction $j$. If the reactant $i$ does not participate in the reaction $j$, then $K_{m,i,j} = 0$. Accordingly, the enzyme-saturation-efficiency criterion for the entire network is defined

$$C_i \ge K_{m,i}^{max}, \quad \forall i \in \mathcal{I}_c. \quad (54)$$

If Eq. (54) is not satisfied for reactant $i$, it implies that at least one of the reactions it participates in is undersaturated. Note that, the saturation-efficiency criterion in this equation is a strong notion of efficiency. It requires reversible enzymes to be half-saturated regardless of the reaction direction or the growth conditions. However, it may be relaxed to describe condition-specific phenotypes by requiring enzymes to be half-saturated only in the direction, in which the respective reaction proceeds.

**Composition of reactants.** We discussed in previous sections how to represent the abiotic constraints with respect to effective quantities. These were defined as weighted averages of the respective quantity for species constituting a reactant. The mole fractions of species were the weights in these definitions. Here, we show how to compute these mole fractions from binding constants for a reactant $\mathcal{A}$. First, we derive expressions to relate hydrogen dissociation and magnesium binding constants at a given ionic strength to those evaluated at the reference condition

(infinite dilution at the same pressure and temperature, where $I \to 0$)

$$K_{0,k} = K_{0,k}^{\circ\circ} \exp[-2(1 - k - z_A)f^\gamma(I^*)], \quad k = 1 \cdots N, \quad (55)$$

$$K_{1,k} = K_{1,k}^{\circ\circ} \exp[4(k + z_A)f^\gamma(I^*)], \quad k = 0 \cdots N, \quad (56)$$

$$K_{2,k} = K_{2,k}^{\circ\circ} \exp[4(2 + k + z_A)f^\gamma(I^*)], \quad k = 0 \cdots N, \quad (57)$$

where $K_{0,k}^{\circ\circ}$, $K_{1,k}^{\circ\circ}$, and $K_{2,k}^{\circ\circ}$ are reference equilibrium constants, and $z_A$ is the charge of the minimum charge state of $\mathcal{A}$ (see "Reactants and species in biochemical reactions" section). Moreover, we define $K_{0,0} := 1$ for notational convenience. Next, we introduce the binding polynomial

$$\mathcal{P} := \sum_{k=0}^N \left(1 + \frac{[Mg]}{K_{1,k}} + \frac{[Mg]^2}{K_{1,k}K_{2,k}}\right)\frac{[H]^k}{\prod_{i=0}^k K_{0,i}} \quad (58)$$

for hydrogen-dissociation and magnesium-binding equilibria given by Eqs. (1)–(3). Finally, we obtain the species mole fractions

$$\rho_{H_k A} = \frac{1}{\mathcal{P}}\frac{[H]^k}{\prod_{i=0}^k K_{0,i}}, \quad k = 0 \cdots N, \quad (59)$$

$$\rho_{MgH_k A} = \frac{1}{\mathcal{P}}\frac{[Mg]}{K_{1,k}}\frac{[H]^k}{\prod_{i=0}^k K_{0,i}}, \quad k = 0 \cdots N, \quad (60)$$

$$\rho_{Mg_2 H_k A} = \frac{1}{\mathcal{P}}\frac{[Mg]^2}{K_{1,k}K_{2,k}}\frac{[H]^k}{\prod_{i=0}^k K_{0,i}}, \quad k = 0 \cdots N. \quad (61)$$

**Formation energy of species from formation energy of minimum charge state.** Before proceeding to calculate the formation energy of species, we provide useful expressions to ascertain the formation energy of species from that of the minimum charge state. The formation energies of all the charge states arising from Eqs. (1)–(3) can be obtained from $\Delta_f G_A^{\circ\circ}$ according to

$$\Delta_f G_{H_k A}^{\circ\circ} = \Delta_f G_A^{\circ\circ} + RT\ln\left(\prod_{i=0}^k K_{0,i}^{\circ\circ}\right), \quad k = 0 \cdots N, \quad (62)$$

$$\Delta_f G_{MgH_k A}^{\circ\circ} = \Delta_f G_{Mg}^{\circ\circ} + \Delta_f G_A^{\circ\circ} + RT\ln\left(K_{1,k}^{\circ\circ}\prod_{i=0}^k K_{0,i}^{\circ\circ}\right), \quad k = 0 \cdots N, \quad (63)$$

$$\Delta_f G_{Mg_2 H_k A}^{\circ\circ} = 2\Delta_f G_{Mg}^{\circ\circ} + \Delta_f G_A^{\circ\circ} + RT\ln\left(K_{1,k}^{\circ\circ}K_{2,k}^{\circ\circ}\prod_{i=0}^k K_{0,i}^{\circ\circ}\right), \quad k = 0 \cdots N. \quad (64)$$

Note that, in deriving these equations, we substituted the reference formation energy of hydrogen ion $\Delta_f G_H^{\circ\circ} = 0$ kJ/mol, and explicitly expressed the reference formation energy of magnesium ion $\Delta_f G_{Mg}^{\circ\circ} = -455.3$ kJ/mol[25].

**Standard transformed Gibbs energy of formation of reactants.** Following the development of Alberty[25], the standard transformed Gibbs energy of formation of the species $A' \in \mathcal{A}$ is written

$$\Delta_f G_{A'}' = \Delta_f G_{A'}'^{\circ} + RT\ln([A']/C^\circ), \quad (65)$$

where $\Delta_f G_{A'}'^{\circ}$ is the standard transformed Gibbs energy of formation evaluated at the same ionic strength, pH, and pMg, accounting for all the nonidealities and external force fields. It can, in turn, be expressed as

$$\Delta_f G_{A'}'^{\circ} = \Delta_f G_{A'}^{\circ} - N_{H,A'}(\Delta_f G_H^{\circ} + RT\ln 10^{-pH}) - N_{Mg,A'}(\Delta_f G_{Mg}^{\circ} + RT\ln 10^{-pMg}) + z_{A'}F\psi. \quad (66)$$

The individual components of $\Delta_f G_{A'}'^{\circ}$ are given by

$$\Delta_f G_{A'}^{\circ} = \Delta_f G_{A'}^{\circ\circ} + RT\ln\gamma_{A'}, \quad (67)$$

$$\Delta_f G_H^{\circ} = \Delta_f G_H^{\circ\circ} + RT\ln\gamma_H, \quad (68)$$

$$\Delta_f G_{Mg}^{\circ} = \Delta_f G_{Mg}^{\circ\circ} + RT\ln\gamma_{Mg}. \quad (69)$$

Here, $F$ is the Faraday constant, $\psi$ is the external potential field at the point where Gibbs energy is evaluated, and the superscript $\circ\circ$ denotes the reference state of Gibbs energy at infinite dilution of the respective species. The standard transformed Gibbs energy of formation of reactant $\mathcal{A}$ is expressed with respect to that of species

$$\exp\left(-\frac{\Delta_f G_{\mathcal{A}}'^{\circ}}{RT}\right) = \sum_{A' \in \mathcal{A}} \exp\left(-\frac{\Delta_f G_{A'}'^{\circ}}{RT}\right), \quad (70)$$

which can be reformulated to

$$\exp\left(-\frac{\Delta_f G_{\mathcal{A}}^{'\circ}}{RT}\right) = \exp\left(-\frac{\Delta_f G_A^{'\circ}}{RT}\right) \sum_{A' \in \mathcal{A}} \exp\left(\frac{\widehat{\Delta_f G_{A'}^{'\circ}}}{RT}\right), \tag{71}$$

where $\Delta_f G_{\mathcal{A}}^{'\circ}$ is the standard transformed Gibbs energy of formation of $\mathcal{A}$, $\Delta_f G_A^{'\circ}$ the standard transformed Gibbs energy of formation of the minimum charge state, and $\widehat{\Delta_f G_{A'}^{'\circ}} := \Delta_f G_A^{'\circ} - \Delta_f G_{A'}^{'\circ}$. Substituting Eqs. (66)–(69) in Eq. (71), we arrive at

$$
\begin{aligned}
\frac{\Delta_f G_{\mathcal{A}}^{'\circ}}{RT} = & \frac{\Delta_f G_A^{\circ\circ}}{RT} + N_{H,A}[\ln(10)\text{pH} - \ln \gamma_H] \\
& - \ln\left\{ \sum_{k=0}^{N} \frac{\gamma_H^k 10^{-k\text{pH}}}{\gamma_{H_k A} \prod_{i=0}^k K_{0,i}^{\circ\circ}} \exp\left[-(z_A+k)\frac{F\psi}{RT}\right] \right. \\
& + \sum_{k=0}^{N} \frac{\gamma_H^k 10^{-k\text{pH}}\gamma_{Mg} 10^{-\text{pMg}}}{\gamma_{MgH_k A} K_{1,k}^{\circ\circ} \prod_{i=0}^k K_{0,i}^{\circ\circ}} \exp\left[-(z_A+k+2)\frac{F\psi}{RT}\right] \\
& \left. + \sum_{k=0}^{N} \frac{\gamma_H^k 10^{-k\text{pH}}\gamma_{Mg}^2 10^{-2\text{pMg}}}{\gamma_{Mg_2 H_k A} K_{1,k}^{\circ\circ} K_{2,k}^{\circ\circ} \prod_{i=0}^k K_{0,i}^{\circ\circ}} \exp\left[-(z_A+k+4)\frac{F\psi}{RT}\right] \right\}
\end{aligned}
\tag{72}
$$

with $N_{H,A}$ and $\Delta_f G_A^{\circ\circ}$ denoting the hydrogen content and reference formation energy of the minimum charge state. Note that, the second and third sums in Eq. (72) correspond to magnesium-bound states, while the first sum arises from protonation states. We estimated the standard transformed Gibbs energy of reactions with and without magnesium-bound states. However, all the results presented in the main text were computed by neglecting the charge states arising from magnesium bindings. As previously stated, we obtained reaction energies using the reference formation energies of species furnished by group-contribution methods[45]. These methods tune their parameters based on expressions that only account for protonation states. Therefore, using these formation energies in Eq. (72) with all the charge states included introduces errors into reaction-energy computations, which are no longer bounded by the guaranteed uncertainty bounds of group-contribution methods.

**Standard transformed Gibbs energy of reaction**. The standard transformed Gibbs energy of reactions and apparent equilibrium constants are obtained from the standard transformed Gibbs energy of formation of reactants

$$\frac{\Delta_r G_j^{'\circ}}{RT} = \sum_{i \in \mathcal{I}_c \cup \mathcal{I}_p} S_{ij} \frac{\Delta_f G_i^{'\circ}}{RT} + \frac{\Delta_r G_{H,j}}{RT} + S_{w,j} \frac{\Delta_f G_w^{'\circ}}{RT}, \tag{73}$$

$$K_j' := \exp\left(-\frac{\Delta_r G_j^{'\circ}}{RT}\right), \tag{74}$$

where $j$ is the reaction index, and

$$\frac{\Delta_r G_{H,j}}{RT} = \tau_{H,j}\left[\ln\left(\frac{\gamma_H(I_c)[H]_c}{\gamma_H(I_p)[H]_p}\right) + \frac{F\Delta\psi}{RT}\right] \tag{75}$$

is the proton motive force associated with reaction $j$ with $\Delta\psi := \psi_c - \psi_p$ the membrane potential. The standard transformed Gibbs energy of reaction $j$ is

$$\frac{\Delta_r G_j'}{RT} = \frac{\Delta_r G_j^{'\circ}}{RT} + \ln \Gamma_j, \quad \Gamma_j := \prod_{i \in \mathcal{I}_c \cup \mathcal{I}_p} \left(\frac{C_i}{C^\circ}\right)^{S_{ij}}, \tag{76}$$

where $\Gamma_j$ is the quotient of reaction $j$. Note that, in this formulation, the concentrations of hydrogen ion and water do not contribute to the reaction quotients nor do their stoichiometric coefficient to $S$. Instead, the concentrations and stoichiometric coefficients of water and hydrogen ion are implicitly accounted for in Eq. (73). The standard transformed Gibbs energy of formation of water is given by

$$\frac{\Delta_f G_w^{'\circ}}{RT} = \frac{\Delta_f G_w^{\circ\circ}}{RT} + \ln \gamma_w - 2\ln \gamma_H + 2\ln(10)\text{pH}. \tag{77}$$

In biological systems, $\gamma_w \approx 1$ because the concentration of reactants are negligible compared to water[25]. In group-contribution methods, $\gamma_w$ is assumed to be a constant and lumped into the reference formation energy $\Delta_f G_w^{\circ\circ}$, and the resulting expression is used to fit the model parameters (i.e., formation energies of species at infinite dilution) to equilibrium data. In this work, we adopt $\Delta_f G_w^{\circ\circ} = -238.7$ kJ/mol[45].

**Growth medium and model parameters**. As previously stated, the concentration of cytoplasmic metabolites, excluding the transmembrane ions, are the only unknowns in Eqs. (26)–(32). The goal in our model is to characterize the CSS for any given growth medium, the composition and parameters of which are fully specified at the outset. The results presented in the main text all describe the metabolic state of E. coli growing in an M9 minimal medium (Supplementary Table S1) during the exponential growth phase. From the composition of the growth medium, we computed the concentrations of all the periplasmic ions, ascertaining effective charges and $I_p$.

Besides the composition of the growth medium, several other parameters must be specified at the outset to determine intracellular concentration ranges using Eqs. (26)–(32). We chose characteristic parameters associated with the exponential growth of E. coli (Supplementary Table S2) to study feasible intracellular concentrations, and how they are restricted by abiotic constraints. All other parameters used in this work to characterize the CSS for all carbon sources are tabulated in Supplementary Data 1–4.

**Characterization of concentration solution space**. In this section, we present two methods for characterizing the CSS constructed by the abiotic constraints outlined previously. First, we compute global bounds on cytoplasmic concentrations and reaction energies using global optimization techniques to determined their respective biologically feasible ranges. The lower bounds are obtained by solving

$$y_{i,\min} := \min_{\mathbf{y} \in \mathcal{Y}} y_i, \quad i \in \mathcal{I}_c, \tag{78}$$

$$\Delta_r G_{j,\min}' := \min_{\mathbf{y} \in \mathcal{Y}} \Delta_r G_j', \quad j = 1 \cdots m, \tag{79}$$

where $\mathcal{Y}$ is given by Eqs. (47) and (48). The upper bounds are similarly determined. Note that, because the CSS is generally nonconvex and can be disconnected, these global bounds only provide intervals, into which feasible concentrations can fall. For example, when $\mathcal{Y}$ is disconnected, $y_i \in [y_{i,\min}, y_{i,\max}]$ does not guarantee that $y \in \mathcal{Y}$. Accordingly, $y_i \in [y_{i,\min}, y_{i,\max}]$ provides a necessary, but not sufficient, condition for $y \in \mathcal{Y}$. Therefore, the feasible concentration ranges defined by global concentration bounds should not be interpreted as intervals, in which the concentration of individual metabolites can freely vary independently of other metabolites.

In the concentration space, the feasible concentration ranges can span several orders of magnitude possibly over a disconnected CSS. Therefore, it is more convenient to perform global optimization computations in the $Y$ space. Second, we compute expectations and standard deviations of concentrations and reaction energies. Given a random variable $X : \mathcal{C} \mapsto \mathbb{R}$ defined over the CSS, the expectation and standard deviation of $X$ are defined as

$$\mathbb{E}(X) := \int_{\mathcal{C}} X d\mu, \tag{80}$$

$$\sigma(X) := \sqrt{\mathbb{E}((X - \hat{X})^2)}, \tag{81}$$

where $\hat{X} := \mathbb{E}(X)$, and $\mu$ is a probability measure of $X$, encapsulating all the intracellular dynamics and the related parameter uncertainties (e.g., kinetic parameters, protein structure, enzyme saturation, and spacial gradients). This probability measure, which is expected to be condition specific, is poorly understood in biological systems. Without any information about the probability measure for the organism of interest, the most natural choice for $\mu$ is the volume measure. However, as previously stated, $\mathcal{C}$ is generally a nonconvex and disconnected set, rendering volume integrals difficult to compute. Therefore, we further simplify our analysis by using a line measure to approximate expectations and standard deviations. We compute these line integrals along random curves generated in the equality-constraint manifold (Supplementary Fig. S5a, b). Since all these computations are performed in the $Y$ space, we first describe briefly the mathematical formulation of the CSS in this space.

Let $\mathcal{M}$ denote the abstract $(n - \ell)$-dimensional manifold in the $Y$ space defined by the linear equalities in Eq. (43) and $\bar{\mathcal{M}}$ its isometric embedding in $\mathbb{R}^n$ furnished by the embedding map $\iota : \mathcal{M} \hookrightarrow Y$ (Supplementary Fig. S5c). We chose this particular embedding to simplify the construction and computation of the foregoing line integrals. Moreover, let $\mathscr{A} := \mathbf{A}^T$, $\mathbf{b} := (\mathbf{f}^{eq})^T$, and

$$\mathcal{F}^i(y) := \exp y^j \mathscr{A}_j^i - b^i. \tag{82}$$

Note that, in previous sections, we did not distinguish between subscripts and superscripts to denote the components of vectors and covectors. Vectors were treated as $n$-tuples, the components of which were represented by subscripts (column vector in matrix form) without reference to the space they were associated with. However, in this section, the components of vectors and covector are denoted by superscripts and subscripts, which are represented by a row and column vectors in matrix form, respectively. The same convention applies in general to covariant and contravariant components of tensors. We also adopt the Einstein summation rule, where repeating a dummy index as a superscript and subscript implies summation over an appropriate range of the index. Assuming that $A$ has a full rank (i.e., rank($\mathbf{A}$) = $\ell$ with $\ell < n$), we have

$$\bar{\mathcal{M}} := \iota(\mathcal{M}) = \{\mathbf{y} \in \mathbb{R}^n : \mathcal{F}(\mathbf{y}) = 0\}. \tag{83}$$

The tangent space at a point $\mathbf{y}$ on $\bar{\mathcal{M}}$ is given by $T_\mathbf{y}\bar{\mathcal{M}} = \ker(D\mathcal{F})$[56]. In matrix form, the derivative of $\mathcal{F}$ can be written $D\mathcal{F} = \mathbf{A}\mathbf{E}$, where $\mathbf{E} := \text{diag}(\exp \mathbf{y})$. Explicit expressions can be derived to construct coordinate charts $\varphi$ for $\mathcal{M}$ using null-space bases of $\mathbf{A}$. Let $\mathbf{N} \in \mathbb{R}^{n \times (n-\ell)}$ be a matrix containing an orthonormal basis of $\ker(\mathbf{A})$. Then, $\Phi : \chi \mapsto \ln(\mathbf{x} + \chi\mathbf{N}^T)$ with $\chi \in \mathbb{R}^{n-\ell}$ contains information about the coordinate chart $\varphi$ at the point $\mathbf{y} = \ln \mathbf{x}$ because $\Phi = \iota \circ \varphi^{-1}$. The components of $\chi$ can be regarded as the coordinates of $\varphi$ at the point $\mathbf{y}$. We can

now define an induced metric[57] for $\mathcal{M}$ according to

$$
\begin{aligned}
g_{kl} &:= g_q\left(\frac{\partial}{\partial\chi^k}, \frac{\partial}{\partial\chi^l}\right) = \bar{g}_y\left(\iota_*\left(\frac{\partial}{\partial\chi^k}\right), \iota_*\left(\frac{\partial}{\partial\chi^l}\right)\right) \\
&= \bar{g}_y\left(\mathcal{G}_k^i\frac{\partial}{\partial y^i}, \mathcal{G}_l^j\frac{\partial}{\partial y^j}\right) = \mathcal{G}_k^i\mathcal{G}_l^j\delta_{ij},
\end{aligned}
\tag{84}
$$

where $\delta$ is the Kronecker delta tensor, $\mathbf{y} = \iota(q)$, and $\bar{g}_{ij} := \bar{g}_y(\partial/\partial y^i, \partial/\partial y^j) = \delta_{ij}$ is the standard Euclidean metric that $Y \equiv \mathbb{R}^n$ is equipped with. Here, $\iota_*$ is the derivative of $\iota$, furnishing $T_y\bar{\mathcal{M}} = \iota_*(T_q\mathcal{M})$, and $\mathcal{G} := D\Phi^{\mathrm{T}} = \mathbf{N}^{\mathrm{T}}\mathbf{E}^{-1}$ in matrix form.

Before generating random curves in $\bar{\mathcal{M}}$, we first require a feasible point of the CSS, preferably far away from all its boundaries, in the $Y$ space. We generate a one-parameter family of interior points by solving

$$
\begin{cases}
\mathbf{y}_q(w) := \arg\ \max\limits_{r,\mathbf{y}}\ r - w\ \|\ \mathbf{y}\ \|_2, \\
\text{s.t.} \\
\exp\mathbf{y}.\mathscr{A} = \mathbf{b}, \\
\mathbf{y}\mathbf{s}_{(j)} + r\ \|\ \mathbf{s}_{(j)}\ \|_2 \le f_j^{\mathrm{ineq}},\ j = 1\cdot\cdot m,
\end{cases}
\tag{85}
$$

where $\mathbf{s}_{(j)}$ denotes the $j$th column of $\mathbf{S}_c$. Note that, to derive Eq. (85), we solved the second equation in the system Eq. (47) for $C_{t,c}^*$, the value of which is substituted in $\mathbf{f}^{\mathrm{eq}}$ and $\mathbf{f}^{\mathrm{ineq}}$, so it is not explicitly included in the system Eq. (85). Observe that, $\mathbf{y}_q(0)$ is the Chebyshev center (ref. [58], Sect. 8.5.1) of the polyhedron defined by the thermodynamic constraints in the $Y$ space that is restricted to $\bar{\mathcal{M}}$. For $w > 0$, the objective function identifies an interior point of this polyhedron that is maximally distanced from all the thermodynamic constraints with respect to the second norm in the $Y$ space, while avoiding arbitrarily large negative values for $\mathbf{y}$. This provides biologically relevant points inside the CSS.

Several approaches can be taken to generate random curves in $\bar{\mathcal{M}}$ from $\mathbf{y}$. Here, we outline a computationally tractable technique. First, a vector $\mathbf{u} = u^k\partial/\partial\chi^k$ is generated in $T_q\mathcal{M}$ using the basis $\partial/\partial\chi^k$ associated with the coordinate chart $\varphi$ discussed before, where the components $u^k \in [-1, 1]$ are random numbers. The embedding of $\mathbf{u}$ in $T_y\bar{\mathcal{M}}$ can be readily obtained according to $\bar{\mathbf{u}} = \iota_*(\mathbf{u}) = \bar{u}^i\partial/\partial y^i$, where $\bar{u}^i = u^k\mathcal{G}_k^i$. Next, a line $\bar{\mathcal{L}}$ is constructed in $T_y\bar{\mathcal{M}}$ that passes through $\mathbf{y}$ along the tangent vector $\bar{\mathbf{u}}$ with the parametric representation $\bar{\mathcal{L}} := \{\mathbf{y}' \in \mathbb{R}^n : \mathbf{y}' = \mathbf{y} + \bar{\mathbf{u}}t, t \in [0, t_{\max}]\}$. Finally, a trajectory $\bar{\mathcal{T}}$ is constructed in $\bar{\mathcal{M}}$ by finding the orthogonal projection of $\bar{\mathcal{L}}$ onto $\bar{\mathcal{M}}$. This is accomplished by solving

$$
\begin{cases}
\mathbf{y}'(t) := \arg\ \min\limits_{\mathbf{y}''\in\mathbb{R}^n}\ \frac{1}{2}\ \|\ \mathbf{y}'' - \mathbf{y} - \bar{\mathbf{u}}t\ \|_2, \\
\text{s.t.} \\
\mathbf{A}\exp\mathbf{y}''^{\mathrm{T}} = \mathbf{b}^{\mathrm{T}}
\end{cases}
\tag{86}
$$

with

$$
\bar{\mathcal{T}} := \{\mathbf{y}' \in \mathbb{R}^n : \mathbf{y}'\ \text{solves Eq. (86)}, t \in [0, t_{\max}]\}.
\tag{87}
$$

The Karush–Kuhn–Tucker (KKT) conditions (ref. [58], Sect. 5.5.3) for Eq. (86) reads

$$
\begin{cases}
\mathbf{y}'' + \boldsymbol{\lambda}^{\mathrm{T}}\mathbf{A}\mathbf{E} = \mathbf{y} + \bar{\mathbf{u}}t, \\
\mathbf{A}\exp\mathbf{y}''^{\mathrm{T}} = \mathbf{b}^{\mathrm{T}},
\end{cases}
\tag{88}
$$

where $\boldsymbol{\lambda}$ is the dual variable corresponding to the equalities in Eq. (86). It is also a column vector associated with the co-normal space $N_\mathbf{y}^*\bar{\mathcal{M}}$. Moreover, $\mathbf{b}$ is associated with the normal space $N_\mathbf{y}\bar{\mathcal{M}}$ and is a row vector. All the other vectors are associated with $T_\mathbf{y}\bar{\mathcal{M}}$ and are row vectors. Differentiating the KKT system Eq. (88) with respect to $t$, we arrive at

$$
\begin{cases}
\dot{\mathbf{y}}'' + \dot{\boldsymbol{\lambda}}^{\mathrm{T}}\mathbf{A}\mathbf{E} + \boldsymbol{\lambda}^{\mathrm{T}}\mathbf{A}\mathcal{E}\dot{\mathbf{y}}''^{\mathrm{T}} = \bar{\mathbf{u}}, \\
\mathbf{A}\mathbf{E}\dot{\mathbf{y}}''^{\mathrm{T}} = \mathbf{0},
\end{cases}
\tag{89}
$$

where overdot denotes derivatives with respect to $t$, and $\mathcal{E} \in \mathbb{R}^{n\times n\times n}$ is defined as

$$
\mathcal{E}_k^{ij} := \begin{cases}
\exp y''^i, & i = j = k, \\
0, & \text{Otherwise.}
\end{cases}
\tag{90}
$$

Equation (89) is integrated with respect to $t$ subject to the initial conditions $\mathbf{y}''(0) = \mathbf{y}$ and $\boldsymbol{\lambda}(0) = \mathbf{0}$ until one of the thermodynamic constraints is violated. Note that, the constraint $\mathbf{y} \le \mathbf{0}$ need not be checked directly because one of the equality constraints in Eq. (89) forces the sum of the metabolite mole fractions to be strictly less than one. As a result, $y''^i = 0$ can never be realized for any metabolites along trajectories constructed by solving Eq. (89). We formulated the global optimization problems Eqs. (78), (79), and (85) in the General Algebraic Modeling System (GAMS)[59] and solved using the global solver BARON[60] equipped with the local nonlinear programming solver CONOPT and the linear programming solver CPLEX. Equation (89) was solved using standard time integrators in MATLAB. All the custom codes used to general the results presented in this paper can be found in Supplementary Code 1.

Once random trajectories $\{\bar{\mathcal{T}}_i\}_1^{n_{\mathrm{traj}}}$ are constructed, the expectation of a random variable $X$, defined by Eq. (80), can be approximated

$$
\mathbb{E}(X) \approx \frac{\sum\limits_{i=1}^{n_{\mathrm{traj}}} \int_{\bar{\mathcal{T}}_i} X\mathrm{d}l}{\sum\limits_{i=1}^{n_{\mathrm{traj}}} \int_{\bar{\mathcal{T}}_i} \mathrm{d}l}
\tag{91}
$$

with the line measure $\mathrm{d}l = \sqrt{\langle\bar{\mathbf{x}}', \bar{\mathbf{x}}'\rangle}\mathrm{d}t = \sqrt{\langle\dot{\mathbf{y}}'\mathbf{E}, \dot{\mathbf{y}}'\mathbf{E}\rangle}\mathrm{d}t = \sqrt{\dot{\mathbf{y}}'\mathbf{E}^2\dot{\mathbf{y}}'^{\mathrm{T}}}\mathrm{d}t$. The standard deviation of $X$ is similarly approximated. The violation probability of thermodynamic constraint associated with reaction $j$ is given by

$$
P_j \approx \frac{n_{\mathrm{viol},j}}{n_{\mathrm{traj}}},\ j = 1\cdot\cdot m,
\tag{92}
$$

where $n_{\mathrm{viol},j}$ is the number of trajectories intersecting the thermodynamic constraint of reaction $j$, and $n_{\mathrm{traj}}$ the total number of randomly generated trajectories. Expectations, standard deviations, and violation probabilities in Fig. 2 are estimated by generating 50,000 trajectories in the CSS.

**Known osmoregulation mechanisms.** Under physiological conditions, there is a balance between reactions donating negative charge to (Fig. 4b, orange arrows) and accepting negative charge from (Fig. 4b, brown arrows) the cytoplasmic fluid, furnishing a stable electroneutral buffer for cellular metabolism. For example, in response to osmotic stress, E. coli imports osmoprotectants (e.g., glycine betaine and trehalose) or $K^+$ from the extracellular environment to modulate its intracellular osmolarity, keeping a fixed osmotic pressure across the cell membrane[61]. The imbalance caused by these charged solutes can be alleviated by upregulating reactions producing counterions or downregulating those consuming counterions.

To maintain electroneutrality, the cell has many degrees of freedom, as to which reactions to upregulate and which ones to downregulate. Importantly, the robustness of E. coli to various environmental stresses suggests that the transcriptional regulatory network performing osmoregulation has evolved to sustain homeostasis with respect to other simultaneous disturbances (e.g., carbon-source changes and pH stress), minimally impacting pathways performing other critical cellular functions.

Many genes in the E. coli genome are involved in osmoregulation. Primary transcriptional regulatory targets include potassium transporters (e.g., trkA and kdpA)[27,61], osmoprotectant transporters (e.g., proV)[27], carbon-source transporters (e.g., crr and ptsG)[31], sodium transporters (e.g., nhaA and nhaB)[26], glutamate transporter (e.g., gadC and gltI)[28], and the ETC (e.g., atpE and cyoD)[29]. Secondary responses affect several intracellular pathways, such as amino-acid biosynthesis, osmoprotectant biosynthesis, central-carbon metabolism, and nucleotide biosynthesis[31].

**Glutamate role in osmoregulation.** To elucidate the glutamate-accumulation mechanism by which electroneutrality and pH homeostasis are achieved, we compared the intrinsic hydrogen and charge consumption (see Eqs. (37) and (36)) of the impacted pathways. Interestingly, all major glutamate-biosynthesis pathways (i.e., GLUDy, GLUSy, and GLUDC) simultaneously consume hydrogen ion and produce negative charge with similarly high intrinsic values ($\Delta_r N_H \approx 1$ mol-H/mol-rxn and $\Delta_r Z \approx -1$ mol-$e$/mol-rxn), whereas glutamate antiporters only produce negative charge ($\Delta_r N_H \approx 0$ mol-H/mol-rxn and $\Delta_r Z \approx -1$ mol-$e$/mol-rxn). These features highlight the importance of glutamate antiporters during phase I when the cytoplasmic solution is alkalinized, allowing the cell to rapidly accumulate glutamate and potassium from the extracellular environment, so as to maintain electroneutrality without a significant disruption to pH homeostasis. Therefore, exporting excess glutamate, possibly through mechanosensitive channels (e.g., mscL)[26], when the cell is under no stress could be part of a robust regulatory network in bacteria, enabling them to create an extracellular glutamate pool, which can be used to flexibly handle various stress conditions causing ion imbalances.

Biochemical reactions can affect the cytoplasmic charge in two different ways, both of which are captured by our definition of $\Delta_r Z$ (see "Abiotic constraints" section). Recall, $\Delta_r Z$ reflects the amount of charge exchange between metabolites and the cytoplasmic fluid for intracellular reactions, while it measures the net charge transfer from the periplasm to the cytoplasm for transport reactions. If the overall cytoplasmic electroneutrality is regarded as a more essential osmoregulation objective than balanced intracellular charge exchanges, then it follows from our analysis that glutamate antiporters play a more significant role in glutamate accumulation during osmoregulation than glutamate-biosynthesis pathways—a conclusion supported by differential expression analyses of RNA-sequencing data.

**Flux distribution.** To compute extrinsic reaction charge, hydrogen, and magnesium consumptions, the flux distribution across the network is required. We determine the flux state of the reduced network during the exponential growth phase using the genome-scale reconstruction model iML1515[15]. To reliably estimate the metabolic fluxes, we identify the parsimonious solution[62] of the resulting FBA model that best matches experimentally measured fluxes[14]. The parsimonious solution, furnished by minimizing a norm of the flux vector, postulates that the optimal flux state minimizes the total protein cost of the network, assuming that all biochemical reactions have the same protein cost. This objective has been shown to be associated with the flux state of several organisms under various conditions[17,38],

and, therefore, is adopted in our study. We formulate the FBA model as a bi-level optimization problem, incorporating the foregoing objectives. To simplify our analysis, we performed several numerical experiments and found that the upper-level objective, measuring the distance between the experimental and predicted fluxes, has the largest sensitivity to the upper bounds on the GLGC and NDPK1 fluxes in our reduced network. Accordingly, we determine the flux state by solving

$$
\begin{cases}
\mathbf{v}_0^* := \arg \min_{\boldsymbol{\theta}, \mathbf{v}} \sum_{j \in \mathcal{J}^{ex}} \left( \frac{v_j - v_j^{ex}}{\sigma_j^{ex}} \right)^2, \\
\text{s.t.} \\
\mathbf{v} \in v_1^*(\boldsymbol{\theta}), \\
\quad
\begin{cases}
v_1^*(\boldsymbol{\theta}) := \arg \min_{\mathbf{v}} \| \mathbf{v} \|_2, \\
\text{s.t.} \\
\mathbf{S}\mathbf{v} = \mathbf{0}, \\
\mathbf{v}^{lb} \le \mathbf{v} \le \mathbf{v}^{ub}(\boldsymbol{\theta}),
\end{cases}
\end{cases}
\tag{93}
$$

where $\mathbf{S}$ is the stoichiometric matrix of iML1515, $\mathcal{J}^{ex}$ the index set of reactions with experimentally measured flux, $\sigma_j^{ex}$ the standard deviation reported for the measurement of flux through reaction $j$, $\mathbf{v}^{lb}$ and $\mathbf{v}^{ub}$ the lower and upper bounds on the flux vector, and $\boldsymbol{\theta}$ a vector containing the upper bounds of GLGC and NDPK1. These upper bounds, which are decision variables in the upper level, are linearly expressed in the respective inequality constraints of the lower level. Thus, the lower-level problem of Eq. (93) is a convex quadratic programming problem, which we solve using the quadratic programing solvers of CPLEX. We solve the upper-level problem using gradient-descent methods.

To evaluate the impact of the confidence intervals $\sigma_j^{ex}$ on the solutions of Eq. (93), we performed flux computations with and without accounting for $\sigma_j^{ex}$ for all four carbons sources and compared the results. Overall, incorporating the confidence intervals into computations did not significantly affect flux values or their directions. However, the flux distribution provided by Eq. (93) for the case where confidence intervals were taken into account generally agreed better (except when pyruvate was the carbon source) with experimental measurements (Supplementary Data 5).

Granted, the procedure that we introduced in this section cannot generally guarantee the correct direction of reactions, for which experimental flux data are unavailable. However, given the size of the reduced network and the number of reactions with flux data, it is unlikely that the enzymatic reactions associated with the reduced network proceed in the opposite direction of their canonical directions determined from the solutions of Eq. (93). Accordingly, the results presented in this work are not expected to be significantly influenced by flux-direction uncertainty.

**Role of phosphate transporters**. We compared the upper bound on the cytoplasmic-phosphate concentration in mutants, where either the *Pit* or *Pst* system is active (Supplementary Fig. S6). We first note that the upper bound on the cytoplasmic-phosphate concentration furnished by the ABC-bases analysis and TMFA coincide at low and high periplasmic concentrations when *Pit* is active. In contrast, when *Pst* is active, the upper bound provided by the ABC-based approach is more restrictive than TMFA. This implies that thermodynamics places a more restrictive constraint on the *Pit* system than the *Pst* system. On the other hand, we observe from Supplementary Fig. S6 that phosphate import through the *Pit* system cannot maintain the cytoplasmic-phosphate concentration at a level that is essential for the growth of *E. coli*. These results indicate that the transition from the *Pit* to *Pst* systems with decreasing $[[\text{pi}]]_p$ is induced by the *Pit* system becoming thermodynamically infeasible in phosphate-limited environments.

**Glycolysis and gluconeogenesis in pH homeostasis**. In the main text, we highlighted an important feature of glycolysis with regards to pH homeostasis. Namely, it produces hydrogen ion at a much lower rate than the ETC, whether it operates in forward direction or reverse (due carbon-source change). Here, we examine this feature in more detail.

We computed the flux state of the reduced network (see "Flux distribution" section) using glucose or pyruvate as the sole carbon source. When the cell grows on glucose, glycolytic reactions are active. However, when the carbon source is changed to pyruvate, glycolysis is reversed and gluconeogenetic reactions become active. We used the computed flux distributions to estimate the extrinsic hydrogen production of glycolysis and gluconeogenesis and found that the rate of hydrogen production remains almost unchanged when the carbon source is altered (Supplementary Fig. S7). We found that gluconeogenesis produces hydrogen ion at a slightly higher rate ($\Delta_r N_H = -29.08$ mmol-H/gDW/h) than glycolysis ($\Delta_r N_H = -21.14$ mmol-H/gDW/h). This may be attributed to lactate dehydrogenase and lactate exporters, which are active only when pyruvate is the carbon source. These reactions consume hydrogen ion, so they offset the slightly larger hydrogen-ion production of gluconeogenesis.

Our results suggest that the evolution of glycolytic and gluconeogenetic pathways enables *E. coli* to robustly restore pH homeostasis when it is simultaneously subject to carbon-source-starvation and acid stress.

**Pathway classification**. We classified reactants and reactions based on the pathway they participate in to contrast the energetics and metabolite distribution across major pathways in the reduced network. The abbreviations used to label these pathways in Supplementary Figs. S2 and S3 are defined in Supplementary Table S3.

**Consistency of the ABCs with metabolomic data**. To quantify the consistency of the ABCs with metabolomic data, we identified the closets point inside the CSS to measured intracellular concentrations[14] when the CSS is defined by (i) thermodynamic constraints alone and (ii) all the ABCs in Eqs. (26)–(31). As expected, the closest point inside the CSS in the first case is better correlated with experimental data than the second case (Supplementary Fig. S12). This is a consequence of the ABC-based analysis being more restrictive than TMFA because there is more freedom in the feasible set of the optimization problem, in which to search for the best fit.

**Confidence intervals for Gibbs energy of reactions**. To impose the thermodynamic constraints given by Eq. (31), we estimated the reference Gibbs energy of formation of reactants using group contribution methods. These formation energies are subject to uncertainties that are bounded by confidence intervals accompanying group-contribution estimates. Confidence intervals for Gibbs energies were not accounted for in Eq. (31). They were also neglected when computing the feasible ranges of metabolite concentrations and transformed Gibbs energy of reactions in Figs. 2 and 3. Confidence intervals can be incorporated into the formulation of thermodynamic constraints according to

$$
\Delta_r G_j^{'\circ} + RT\ln\Gamma_j + \delta_j \le 0, \; j = 1 \cdots m, \tag{94}
$$

$$
-\delta_r G_j^{'\circ} \le \delta_j \le \delta_r G_j^{'\circ}, \; j = 1 \cdots m. \tag{95}
$$

Here, $\delta_r G_j^{'\circ}$ is the standard deviation of errors associated with group-contribution computation of the standard transformed Gibbs energy of reaction $j$[10], and $\delta_j$ are auxiliary variables introduced into the ABC-based framework to account for these errors.

Inclusion of confidence intervals in the ABC-based analysis relaxes the feasible ranges of metabolite concentrations and transformed Gibbs energy of reactions. Specifically, it increases the upper bounds and decreases the lower bounds. We repeated the computations of the feasible ranges reported in Figs. 2 and 3 with confidence intervals and observed no significant difference in the upper and lower bounds. Accordingly, all the phenotypic consequences of the ABCs discussed in the main text remained unchanged. For example, the ABC-based analysis predicts three more irreversible reactions (i.e., GLUN, GLUSy, NH4tpp) than TMFA when confidence intervals are included (Supplementary Fig. S13). We also compared the extent, to which the inclusion of confidence intervals could relax the upper and lower bounds provided by the ABC-based analysis and TMFA and found that the predictions of TMFA were more sensitive to confidence intervals than those of the ABC-based analysis (Supplementary Table S4).

**Reporting summary**. Further information on research design is available in the Nature Research Reporting Summary linked to this article.

## Data availability
All data generated or analyzed during this study are included in this published article and its Supplementary Information files.

## Code availability
All the codes, their description, and data used for analysis are provided in Supplementary Information files.

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

## Acknowledgements

We thank Jared Broddrick's for his valuable comments on the manuscript. We would like to thank Sharon Grubner and Jonathan Hsu. This work was funded by the Novo Nordisk Foundation (Grant Number NNF10CC1016517), the National Institutes of Health (Grant Number GM057089), and the Institute for Systems Biology's Translational Research Fellows Program (J.T.Y.).

## Author contributions

Conceptualization, A.A., D.C.Z., J.T.Y., and B.O.P.; methodology, A.A.; validation A.A.; formal analysis A.A.; investigation, A.A., D.C.Z., and J.T.Y.; writing—original draft, A.A., B.O.P., and D.C.Z.; writing—review and editing, A.A., D.C.Z., J.T.Y., and B.O.P.; funding acquisition, B.O.P.; resources, B.O.P.; supervision, B.O.P.

## Competing interests

The authors declare no competing interests.
