## [Peer Review File · Nature Communications]

Reviewers' Comments:

Reviewer #1:

Remarks to the Author:

In their manuscript, Akbari et al. list 10 physicochemical constraints for the concentrations of intracellular metabolites (which they call abiotic constraints or ABCs) and explore their consequences on cellular metabolism and its regulation. They do this using an extension of flux balance analysis (FBA) on a reduced model of *E. coli* metabolism. They find that (1) experimentally measured metabolite concentrations are consistent with the constraints; (2) high-affinity phosphate transporters in *E. coli* differ in terms of thermodynamics, explaining why evolution favored two distinct systems; (3) constraint-based reasoning can rationalize part of the transcriptional response to osmotic and acid stress.

The authors rightly emphasize that physicochemical constraints are important in physiology and hence evolution. Their analysis is an important first step toward systematically elucidating these connections on a network level. However, their manuscript suffers from some overgeneralizations that should be toned down (see below).

OVERGENERALIZATIONS

1. Abstract and L.229-242: "(i) abiotic constraints drive the evolution of transport systems, such as..." – the single "such as..." case is really all that the analysis covers, and this case was driven (apparently) only by thermodynamics, not by ABCs in general.
2. Figure 4 legend: "The ABC-based analysis reveals the biophysical mechanisms underpinning the metabolic functions of *E. coli*" – this seems to be an overstatement.
3. L.297 "These observations allowed us to decipher the fundamental principles ..." – same here. Please be specific about what and to what extent can be revealed and deciphered.
4. L.526 (final sentence) "the ABC-based analysis shows that biological interactions with the inorganic world are foundational to the evolution and function of cells" – the notion that physicochemical constraints have shaped the evolution and function of cells is the basis of the science of biochemistry, and not something that the current analysis discovered.

OTHER SPECIFIC COMMENTS

5. Abstract: "We show that the quantitative metabolomes allowed by the abiotic constraints are consistent with ... isotope-labeling data" (repeated in Discussion L.438). In the Results, this claim is made in L.182 ("(i) They are consistent with previous reports based on ²H and ¹³C metabolic flux analysis for glycolysis [20] and TCA cycle [21, 22]."). However, no details are given. What is meant by consistency here, and how was it quantified?
6. L.53 – this sentence summarizes what the authors actually did in VERY abstract, high-level terms. To allow the reader to appreciate what was done without first reading the Methods, this description should be expanded significantly.
7. L.121-125 (discussion of cofactor concentrations) – cofactors such as ATP/ADP provide thermodynamic driving forces. To this reviewer, it is thus thermodynamics that likely explain why "their concentrations may be driven towards the CSS boundaries". Please discuss this possibility.
8. L.140 (and paraphrased in L.191) "We found 14 undersaturated reactions ($C < K_{max}$) from central-carbon pathways (e.g., PGI, ICDH_{yr}, TALA, and AKGDH), indicating that enzyme efficiency is not a fundamental cellular objective or constraint" – this is a bold statement that requires some discussion.

From finding some counterexamples the authors can certainly derive that enzyme efficiency is not a fundamental constraint (which should be obvious), but it likely is one of several competing cellular objectives (see, e.g., <https://www.biorxiv.org/content/10.1101/128009v1>).

9. L.159 "violation probabilities" – please define these, referring to Methods is not sufficient here.

10. L.268 "The reactions having the highest rate of exchange with the cytoplasmic fluid represent the key players in maintaining hydrogen and metal-ion concentrations. Thus, they are prime targets for evolving transcriptional regulatory mechanisms" – not necessarily, regulation could instead occur at an upstream reaction.

11. L.287 "primary response genes" and L.292 "secondary regulatory targets" – how are these defined and identified?

12. L.314 (and following paragraphs) – It is not immediately obvious why this analysis of glutamate production and import requires the full ABC framework; this should be discussed.

13. L.363 "We identified 41 differentially expressed genes with the same regulatory targets" – how were the targets identified?

14. Constraints (Methods, p.29-31): The constraints should be discussed and justified in more detail, in particular:

a. Eq.(29) – why do the authors expect the total molar concentration of cytoplasmic + periplasmic metabolites + ions to be bounded, rather than, e.g., their volume or mass concentration?

b. Eq.(31) – why is this thermodynamic constraint referred to in the text as the "mass action principle" (L.730)?

c. Eq.(32) – this is called a "scaling expression", but I don't think that is a well-defined term in this context; it should be explained in more detail.

d. Eq. (42) – the set is defined by "Eqs. (26) – (31) hold". This excludes Eq. (32), the approximate saturation constraint; that should be made explicit and it should be justified why Eq. (32) is still considered a constraint.

15. L. 866 "Reversible reactions are required to be half-saturated or more in both directions to be considered efficient." – This sentence requires some justification. A reversible reaction is generally not used in both direction in a given steady-state condition, so it is not clear why efficiency requires product saturation. It appears that here and when taking the maximum across K_m values for one metabolite (Eq. (54)), the authors aim for a network-based (not phenotype-based) characterization that ignores condition-specific differences. This is a defensible approximation, but it has to be discussed as such.

MINOR COMMENTS

16. The authors should carefully review their use of commas and hyphens (e.g., "...-consumption") – a large fraction of both are placed wrongly and should be removed.

17. "standard transformed energy", "standard transformed formation energy" – please replace with "standard transformed Gibbs energy" and "standard transformed Gibbs energy of formation", respectively, throughout for clarity

18. Abstract: "(ii) network-wide characterization of charge-, hydrogen- and magnesium-related constraints shape transcriptional regulatory responses to osmotic stress;" – the characterization only shapes our thoughts, please rephrase.

19. L.52 "maybe" -> "may be"

20. Fig. 2 legend: "(A) Pearson correlation of...C_{ex} ... and .. C_{cm}" – the figure plots C_{ex} vs. C_{cm}, not their Pearson correlation.

21. L.179 – give units for the Gibbs free energy

22. Figure 2 "The ABCs are consistent with metabolomic and isotope-labeling data" – where is the isotope labeling data in this figure?

23. L.240, L.1153 "forgoing" -> "foregoing"

24. L.381 – please define extrinsic hydrogen consumption

25. L.519 "These constraints can quantify perturbations in charge, pH, and pMg homeostasis induced by biochemical reactions" – constraints do not quantify perturbations (though they may help to do so).

26. L.596 – remove one word in "the this"

27. L. 850 – It should be clarified that the specific form of the rate law arises only under certain assumptions about the order of individual reaction steps

28. L.857 "both efficiencies" – efficiencies are only defined in the following sentences, please restructure.

29. L.1148 – please cite Holzhütter 2004 for parsimonious FBA (doi: 10.1111/j.1432-1033.2004.04213.x.)

Signed review:
Martin Lercher

Reviewer #2:

Remarks to the Author:

The study of Akbari et al. provides formulation of abiotic (so-called ABC) constraints in the constraint-based modelling framework and uses the resulting nonlinear and nonconvex problem to make statements about: (i) how charge-, hydrogen-, and magnesium-constraints shape transcriptional regulatory responses to stresses and (ii) evolution of high-affinity phosphate transporters. The work is of potential interest as it bridges a gap between constraint-based approaches that can make predictions and inferences of flux but not concentration states of an investigated metabolic network. The abiotic constraints can be seen as an extension of thermodynamic metabolic flux analysis (TMFA) and include ten different classes of abiotic constraints that lead to a nonconvex and disconnected set of feasible solutions (for metabolite concentrations). As a result of the complexity of the underlying problem in ABC-based analysis, a number of questions arise that require careful consideration.

Major

1. Given that the fundamental constraints must hold for having any meaningful predictions from the constraint-based modelling framework, it is surprising that one of the main findings is – trivial – that measured metabolomes respect these constraints.
2. To "ascertain" / predict a concentration state, the authors minimize the distance between predicted metabolite concentrations and measured concentrations. In this respect, the presented findings from the ABC-based analysis are not and cannot be considered as predictions, but rather fits. Hence, it is not surprising that a model of high quality along with meaningful constraints lead to good fits (high Pearson correlation coefficients reported).
3. Given point 2, above, the ABC-based analysis, like TMFA, does not provide the means for prediction

of concentrations – particularly given the order or magnitude differences in the ranges that the concentration of a particular metabolite can take (Fig. 2). The ABC-based analysis may result in more restricted feasible concentration ranges. Here, it would be better to quantify the reduction in predicted feasible concentration ranges between TMFA and ABC-based analysis (Fig. 3). In addition, it would be good to quantify the Pearson correlation with TMFA constraints when concentrations are fitted – for comparison. At this point, I wonder if the additional two irreversible reactions found by the ABC-based analysis are due to artefacts mentioned in the points 6 – 9, below.

4. The ABC-based approach is applied to a core model of *E. coli* consisting of 78 reactions, 72 cytosolic and 11 periplasmic metabolites. The reason for selecting this model is that the considered metabolites already cover 90% of the observed metabolome (by mole). It is best to be realistic and specify that the approach is applied to a small network due to the computational complexity of the programs that are formulated and some of the developed approaches.

5. A sampling approach was developed to obtain estimates for the expected concentrations of metabolites. The sampling approach (Eqs. (85) – (86)) relies on generation of random curves in a embedding of a manifold generated by the equalities in the ABC-based analysis. Interestingly, in a recently published companion work by Akbari & Palsson (2020), the proposed sampling procedure failed to identify the feasible concentration region a pathway, like glycolysis, which is considerably smaller than the network analysed here. It would be interesting to show if and to what extent the derived estimates for the expected concentrations are biased.

6. The work also relies on determining a flux state and respective inputs. To this end, fluxes were fitted by a bilevel optimization problem (Eq. (93)). The experimentally determined fluxes come with confidence intervals which should be factored in the flux fitting (e.g. via Mahalanobis distance) for unbiased characterization of flux space.

7. Related to point 6, the bilevel program provides a flux estimate which minimizes the second norm of the flux distribution. Are any of the fluxes estimated in such a way zero? If so, based on Eqs. (50) – (52), some of the concentrations of the substrates must be zero or the enzyme should be either not expressed or inactive. Further elaboration of the flux state is needed to justify its usage in the downstream analyses.

8. Related to point 6 and 7, the reaction direction is fixed to the net flux in the flux from the bilevel optimization. Does the bilevel optimization not result in alternative solutions? If there is any alternative solution, i.e. flux distribution with different sign pattern, the downstream analysis and resulting findings will be undoubtedly biased. Further support for the decision to fix reaction directionality is needed.

9. Similar issues due to alternative optima / concentrations closest to measured ones in Gerosa et al. also bring into question the conclusions from the thermodynamic bottleneck analysis.

10. How sensitive are the findings in Fig. 3 C based on the imposed constraints on the periplasmic concentration of phosphate? Where do these numbers come from?

11. To make sense of the claims in the transcriptional response to osmotic and acid stress, it is important to show that there are statistically significant association between up/down-regulation of transcripts and changes on the level of flux. Currently, the last two sections read like a series of observations which are linked, to some extent, to previously published results, but do not provide direct mechanistic explanation for the transcriptional changes.

12. The findings are based on transformed Gibbs free energy of reactions without magnesium-bound states, with the reasoning that these will be erroneous due to the usage of reference formation energies from the group contribution method. The group contribution method also provides confidence intervals for the estimates, which are usually incorporated in TMFA. The formulation of the ABC-based analysis should be updated to consider these uncertainties and re-evaluate the findings.

13. It would be illuminating to specify the reasons why Eqs. (47) – (48) are more computationally tractable than Eqs. (43) and (46).

Minor

It is essential to specify that the transformed Gibbs free energy is used since pH is fixed a priori to the simulations.

Reviewer #1 (Expertise: Metabolic modeling):

In their manuscript, Akbari et al. list 10 physicochemical constraints for the concentrations of intracellular metabolites (which they call abiotic constraints or ABCs) and explore their consequences on cellular metabolism and its regulation. They do this using an extension of flux balance analysis (FBA) on a reduced model of *E. coli* metabolism. They find that (1) experimentally measured metabolite concentrations are consistent with the constraints; (2) high-affinity phosphate transporters in *E. coli* differ in terms of thermodynamics, explaining why evolution favored two distinct systems; (3) constraint-based reasoning can rationalize part of the transcriptional response to osmotic and acid stress.

The authors rightly emphasize that physicochemical constraints are important in physiology and hence evolution. Their analysis is an important first step toward systematically elucidating these connections on a network level. However, their manuscript suffers from some overgeneralizations that should be toned down (see below).

OVERGENERALIZATIONS

1. Abstract and L.229-242: “(i) abiotic constraints drive the evolution of transport systems, such as...” – the single “such as...” case is really all that the analysis covers, and this case was driven (apparently) only by thermodynamics, not by ABCs in general.

We have rephrased this sentence in abstract as suggested by the reviewer, making it specific to phosphate transporters.

2. Figure 4 legend: “The ABC-based analysis reveals the biophysical mechanisms underpinning the metabolic functions of *E. coli*” – this seems to be an overstatement.

We have rephrased this sentence as suggested by the reviewer (highlighted in the caption of Fig.4).

3. L.297 “These observations allowed us to decipher the fundamental principles ...” – same here. Please be specific about what and to what extent can be revealed and deciphered.

We have rephrased this sentence as suggested by the reviewer (highlighted on page 14). Please note that we started this paragraph with a general statement about the connection between reaction charge consumption and transcriptional regulatory response to osmotic stress. However, towards the end of the paragraph, we specifically discuss how reaction charge consumption could be linked to the role of glutamate antiporters in osmoregulation.

4. L.526 (final sentence) “the ABC-based analysis shows that biological interactions with the inorganic world are foundational to the evolution and function of cells” – the notion that physicochemical constraints have shaped the evolution and function of cells is the basis of the science of biochemistry, and not something that the current analysis discovered.

We have rewritten the last paragraph in the revision (highlighted on page 21).

OTHER SPECIFIC COMMENTS

5. Abstract: “We show that the quantitative metabolomes allowed by the abiotic constraints are consistent with ... isotope-labeling data” (repeated in Discussion L.438). In the Results, this claim is made

in L.182 (“(i) They are consistent with previous reports based on 2H and 13C metabolic flux analysis for glycolysis [20] and TCA cycle [21, 22].”). However, no details are given. What is meant by consistency here, and how was it quantified?

We used two metrics to quantify the consistency between the ABCs and experimental data, namely, Pearson’s correlation coefficient R and root mean square deviation RMSD in Fig. 1A. Here, consistency refers to the fact that metabolomic data across all carbon sources lie inside the concentration solution space (CSS) as indicated by the foregoing two metrics. Please note that this consistency should not be taken for granted. Recalling that the CSS is constructed by intersecting an affine space (defined by a set of equality constraints) with a nonconvex set (defined by the thermodynamic constraints), it is unlikely that any arbitrary or inconsistent set of constraints/parameters would furnish a nonempty CSS in a ~ 70 -dimensional space, much less one that would contain concentration data across four carbon sources. We should also emphasize that, previous constraint based models like NET or TMFA, set upper and lower concentration bounds at the beginning ($10^{-5} M \leq C \leq 0.02 M$) before computing feasible ranges. Here, we do not impose any upper or lower bounds on intracellular concentrations at the outset, and all the feasible ranges computed in our analysis arise from the ABCs alone.

6. L.53 – this sentence summarizes what the authors actually did in VERY abstract, high-level terms. To allow the reader to appreciate what was done without first reading the Methods, this description should be expanded significantly.

We have rewritten and expanded this paragraph as suggested by the reviewer (highlighted on page 3).

7. L.121-125 (discussion of cofactor concentrations) – cofactors such as ATP/ADP provide thermodynamic driving forces. To this reviewer, it is thus thermodynamics that likely explain why “their concentrations may be driven towards the CSS boundaries”. Please discuss this possibility.

It is true that cofactors provide thermodynamic driving force. However, as our results in Figs. 2C, 4A and also those previously reported (Park et al., Nat. Chem. Biol., 2019) indicate, intracellular concentrations are such that a number of reactions in central carbon metabolism remain close to their equilibrium. Whether or not a reaction is co-factor driven, thermodynamic constraints restrict the concentrations of the metabolites participating in that reaction only when it is close to its equilibriums. The farther away from the equilibrium, the more freedom concentrations have to vary. One should also note that thermodynamics place constraints on concentration ratios and not their absolute values. So, even if a reaction of the form $S + X_{rd} \rightleftharpoons P + X_{ox}$ (S is the substrate, P the product, X_{rd} and X_{ox} are the reduced and oxidized forms of the cofactor X) is near its equilibrium, thermodynamics can reduce the degree of freedom from 4 to 3 at most. Therefore, just from the fact that a given reaction of a network is close to its equilibrium, it is not straightforward to deduce which metabolite concentration is restricted by thermodynamics. It is therefore important to determine feasible concentration range by accounting for all the constraints imposed on the network. As our results in Fig. 2B suggest, the concentration of certain metabolites (including most of the cofactors) can still vary by several orders of magnitude without violating any constraints.

Another point we would like to clarify is that there is a difference between a reaction being close to its equilibrium at a particular point inside the CSS and one that is forced to remain near equilibrium because the constraints on the network restrict $\Delta_r G'$ to vary in a narrow range around zero for all feasible concentration states. We may attribute the fact that concentrations are close to the boundaries

of the CSS solely to the constraints (thermodynamics for example) imposed on the network in the second case, but not in the first. In this manuscript, all the near equilibrium reactions that we discuss are of the first kind.

We, therefore, believe that the concentration of cofactors is determined mostly by other important constraints that we did not include in the ABC-based analysis due to computational challenges, namely global energy/redox balance and kinetic constraints. These additional constraints can be accounted for only in a full dynamic model of the cell or by coupling the concentration and flux states within a constraint-based modeling framework. We believe that these constraints could significantly restrict the concentration of cofactors (of course, under a specific growth or stress condition) because of the role they play in the catalytic activity of enzymes. Therefore, even if a reaction is far from equilibrium, the cell must maintain the concentration of cofactors close to some steady state values to maintain a balanced flux distribution through the network, ensuring a homeostatic energy/redox balance and an optimal growth rate as result.

8. L.140 (and paraphrased in L.191) “We found 14 undersaturated reactions ($C < K_{max}$) from central-carbon pathways (e.g., PGI, ICDHyr, TALA, and AKGDH), indicating that enzyme efficiency is not a fundamental cellular objective or constraint” – this is a bold statement that requires some discussion. From finding some counterexamples the authors can certainly derive that enzyme efficiency is not a fundamental constraint (which should be obvious), but it likely is one of several competing cellular objectives (see, e.g., <https://www.biorxiv.org/content/10.1101/128009v1>).

We agree with the reviewer that enzyme saturation efficiency could be one of several objectives that the cell must fulfill simultaneously to maintain homeostasis under a particular set of growth conditions. The message we are trying to convey here is that even when the cell is under no apparent stress and growing in a minimal medium, there are still a number of enzymes in the core metabolism that are less than half-saturated. It is in this sense that these enzymes are not efficient, and therefore should not be imposed as a hard constraint or by a single objective function in constraint-based models. We have added a sentence to this paragraph to clarify this point (highlighted on page 8).

9. L.159 “violation probabilities” – please define these, referring to Methods is not sufficient here.

Violation probabilities are defined visually in Fig.1C beside the Method section. Their definition is also explicitly stated in the caption of Fig.1C. We have corrected this sentence, now referring the reader to Fig.1C and its caption for the definition of violation probabilities (highlighted on page 8).

10. L.268 “The reactions having the highest rate of exchange with the cytoplasmic fluid represent the key players in maintaining hydrogen and metal-ion concentrations. Thus, they are prime targets for evolving transcriptional regulatory mechanisms” – not necessarily, regulation could instead occur at an upstream reaction.

This sentence does not exclude the possibility that upstream reactions could also be regulatory targets. What is meant by this sentence is that those reactions that have the highest rate of ion/charge exchange with the cytoplasmic fluid could potentially be targets for transcriptional regulatory networks because they induce larger perturbations if regulated. Therefore, the remainder of discussion in the subsequent sections is going to focus on these potential targets.

11. L.287 “primary response genes” and L.292 “secondary regulatory targets” – how are these defined and identified?

Primary-response genes are those associated with membrane proteins that are directly involved in stress response to restore ion and pH homeostasis, in the case of osmotic and acid stress for example. ETC proteins, potassium, and osmoprotectant transporters are among the primary targets. The secondary-response genes correspond to intracellular reactions that are affected as a consequence of the perturbations induced by the primary response genes. In the case of acid stress for example, where glutamate biosynthetic pathways and antiporters are upregulated, the regulations of TCA and glycolysis reactions are viewed as secondary responses, the purpose of which is to direct material flow towards these activated pathways. We have adopted these terminologies from the literature on acid and osmotic stress regulations.

12. L.314 (and following paragraphs) – It is not immediately obvious why this analysis of glutamate production and import requires the full ABC framework; this should be discussed.

The two constraints that are relevant to glutamate biosynthesis/import the most are charge balance and ion bindings. In this paper, our focus is solely on a constraint-based analysis of intracellular concentrations. As such, the purpose of Sections 2.5 and 2.6 is to present two case studies to point out the significance of the additional constraints that we introduced in the ABC-based analysis for the interpretation of transcriptional regulatory responses. The ultimate goal is to integrate these additional constraints with those on reaction fluxes and protein expressions to be able to predict transcriptional responses to stress conditions. We have added a sentence at the beginning of this paragraph to make this point clear (highlighted on page 14).

13. L.363 “We identified 41 differentially expressed genes with the same regulatory targets” – how were the targets identified?

Among the genes associated with the reduced network, those whose expression levels changed by more than 1.4 fold within an hour of pH downshift were considered differentially expressed. This is the criterion used by the authors of the paper, from which expression data were taken. This criterion is stated in the same sentence in parenthesis.

14. Constraints (Methods, p.29-31): The constraints should be discussed and justified in more detail, in particular:

a. Eq.(29) – why do the authors expect the total molar concentration of cytoplasmic + periplasmic metabolites + ions to be bounded, rather than, e.g., their volume or mass concentration?

We impose the boundedness of the cytoplasm by Eq. (29). This is the boundedness of the total volume or total mass (whichever we assume to be bounded, the boundedness of the other follows because the total density of the cytoplasmic fluid can be reasonably assumed to be a finite number). The boundedness of the mass concentrations of individual reactants, and consequently their sum, in turn follows from the boundedness of the total mass. Moreover, mass and molar concentrations are related to one another through a finite multiplicative constant (i.e. molecular weight). Therefore, the bounded sum of molar concentrations and mass concentrations are the same concepts.

Please note that the total cytoplasmic concentration $C_{t,c}$ is not a fixed parameter that is provided at the outset. It is a variable, which is determined as part of the solution.

b. Eq.(31) – why is this thermodynamic constraint referred to in the text as the “mass action principle” (L.730)?

The mass action principle ($\Gamma \leq K'_{eq}$) is a consequence of the second law of thermodynamics and the dissipative structure of chemical reactive systems, as we pointed out in Methods: Abiotic Constraints. In the literature, it is sometimes referred to as the second law constraint or thermodynamic constraint. In this paper, the terms mass-action principle and thermodynamic constraints have been interchangeably used.

c. Eq.(32) – this is called a “scaling expression”, but I don’t think that is a well-defined term in this context; it should be explained in more detail.

d. Eq. (42) – the set is defined by “Eqs. (26) – (31) hold”. This excludes Eq. (32), the approximate saturation constraint; that should be made explicit and it should be justified why Eq. (32) is still considered a constraint.

What we mean by scaling expression is that we did not generally impose it as a hard constraint, but used it to evaluate the saturation level of enzyme and how well it describes the functional state of the cell by comparing metabolomic data with $K_{m,max}$ values. In Fig.2B, however, feasible concentration ranges for two case are shown, namely when it is imposed as a constraint (in which case the dark gray bars are infeasible because they violate the enzyme saturation constraint) and when it is not (in which case the dark gray bars are part of the feasible range). We have added a paragraph to clarify this point (highlighted on page 31).

15. L. 866 “Reversible reactions are required to be half-saturated or more in both directions to be considered efficient.” – This sentence requires some justification. A reversible reaction is generally not used in both direction in a given steady-state condition, so it is not clear why efficiency requires product saturation. It appears that here and when taking the maximum across K_m values for one metabolite (Eq. (54)), the authors aim for a network-based (not phenotype-based) characterization that ignores condition-specific differences. This is a defensible approximation, but it has to be discussed as such.

This definition is particularly relevant to central carbon reactions where they undergo direction reversal upon carbon-source alteration. In such cases, our definition of enzyme-saturation efficiency furnishes a more robust notion of efficiency by requiring an enzyme to be both substrate and product half-saturated so that it remains efficient regardless of the conditions the cell is growing in. Of course, as the reviewer pointed out, a more relaxed notion of efficiency may be applied to describe condition-specific phenotypes. We have added a discussion at the end of this section to clarify this point (highlighted on page 37).

MINOR COMMENTS

16. The authors should carefully review their use of commas and hyphens (e.g., “...-consumption”) – a large fraction of both are placed wrongly and should be removed.

We have revised the manuscript and corrected punctuation errors.

17. “standard transformed energy”, “standard transformed formation energy” – please replace with “standard transformed Gibbs energy” and “standard transformed Gibbs energy of formation”, respectively, throughout for clarity

We have revised the manuscript and corrected all the inconsistencies pointed out by the reviewer.

18. Abstract: “(ii) network-wide characterization of charge-, hydrogen- and magnesium-related constraints shape transcriptional regulatory responses to osmotic stress;” – the characterization only shapes our thoughts, please rephrase.

We have corrected this sentence in the revised manuscript (highlighted on page 1).

19. L.52 “maybe” -> “may be”

We have corrected this error in the revised manuscript.

20. Fig. 2 legend: “(A) Pearson correlation of...C_{ex} ... and .. C_{cm}” – the figure plots C_{ex} vs. C_{cm}, not their Pearson correlation.

We have changed the caption of Fig.2A to “Comparison of ...” in the revised manuscript (highlighted on page 6).

21. L.179 – give units for the Gibbs free energy

We have applied this comment in the revised manuscript (highlighted on page 9).

22. Figure 2 “The ABCs are consistent with metabolomic and isotope-labeling data” – where is the isotope labeling data in this figure?

Here, consistency with isotope-labeling data refers to the fact that the near equilibrium reactions in Figs. 2C and 4A agree with those reported in the literature based on isotope-labeling data. The data are not shown here, but this point is discussed in the text. Nevertheless, we have removed ‘isotope-labeling data’ from the caption to avoid confusion.

23. L.240, L.1153 “forgoing” -> “foregoing”

We have corrected this error in the revised manuscript (highlighted in page 10,29,49).

24. L.381 – please define extrinsic hydrogen consumption

Intrinsic and extrinsic quantities are defined in Methods: Abiotic Constraints on page 32. We have added a note the first time these quantities are mentioned in the main text in parenthesis, referring the reader to the Method section for the definition (highlighted on page 15).

25. L.519 “These constraints can quantify perturbations in charge, pH, and pMg homeostasis induced by biochemical reactions” – constraints do not quantify perturbations (though they may help to do so).

We have applied this comment in the revised manuscript (highlighted on page 21).

26. L.596 – remove one word in “the this”

We have corrected this error in the revised manuscript.

27. L. 850 – It should be clarified that the specific form of the rate law arises only under certain assumptions about the order of individual reaction steps

We have added a sentence to clarify this point by the reviewer (highlighted on page 36).

28. L.857 “both efficiencies” – efficiencies are only defined in the following sentences, please restructure.

We have introduced the enzyme-saturation and thermodynamic efficiencies prior to this sentence in the revised manuscript (highlighted on page 36).

29. L.1148 – please cite Holzhütter 2004 for parsimonious FBA (doi: 10.1111/j.1432-1033.2004.04213.x.)

We have cited this reference in the revised manuscript.

Signed review:

Martin Lercher

Reviewer #2 (Expertise: Metabolic Modeling):

The study of Akbari et al. provides formulation of abiotic (so-called ABC) constraints in the constraint-based modelling framework and uses the resulting nonlinear and nonconvex problem to make statements about: (i) how charge-, hydrogen-, and magnesium-constraints shape transcriptional regulatory responses to stresses and (ii) evolution of high-affinity phosphate transporters. The work is of potential interest as it bridges a gap between constraint-based approaches that can make predictions and inferences of flux but not concentration states of an investigated metabolic network.

The abiotic constraints can be seen as an extension of thermodynamic metabolic flux analysis (TMFA) and include ten different classes of abiotic constraints that lead to a nonconvex and disconnected set of feasible solutions (for metabolite concentrations). As a result of the complexity of the underlying problem in ABC-based analysis, a number of questions arise that require careful consideration.

Major

1. Given that the fundamental constraints must hold for having any meaningful predictions from the constraint-based modelling framework, it is surprising that one of the main findings is – trivial – that measured metabolomes respect these constraints.

The consistency between the ABCs and metabolomic data is presented here more as a sanity check than a main finding. In the ABC-approach, a complex set of biophysical constraints that rely on a variety of disparate datasets for parameters are used to quantify intracellular concentrations, the phenotypic consequences of which are not obvious to deduce without performing concrete computations. We believe it is important to present these consistency results at the beginning to ensure the reader of the validity of these constraints before proceeding to the main results concerning the phenotypic implications of the ABCs for the concentration state of the cell. We also believe that such consistency results are neither self-evident nor guaranteed. As has been shown in recent studies [1], when modeling biological systems that involve multipart mechanistic sub-models and heterogeneous datasets of parameters, there is no guarantee that the underlying biophysical models and datasets work together coherently to yield consistent results across different conditions.

2. To “ascertain” / predict a concentration state, the authors minimize the distance between predicted metabolite concentrations and measured concentrations. In this respect, the presented findings from the ABC-based analysis are not and cannot be considered as predictions, but rather fits. Hence, it is not surprising that a model of high quality along with meaningful constraints lead to good fits (high Pearson correlation coefficients reported).

Similarly to TMFA, our approach does not predict a specific concentration state. However, it predicts upper and lower bounds. To avoid confusion, we have removed the term ‘predict’ when referring to a particular concentration state from the manuscript (e.g. predictions->results highlighted on page 6). However, the fact that we are able to identify a solution inside the CSS that agrees well with experimental data should not be taken for granted. The CSS is constructed by intersecting an affine space (defined by a set of equality constraints) with a nonconvex set (defined by the thermodynamic constraints). This concentration set is significantly more restricted than the set defined by thermodynamic constraints alone. It is, therefore, unlikely that any arbitrary or inconsistent set of constraints/parameters would furnish a nonempty CSS in a ~70-dimensional space, much less one that

would contain concentration data across four carbon sources. We should also emphasize that, previous constraint-based models like NET or TMFA, set upper and lower concentration bounds at the beginning ($10^{-5} M \leq C \leq 0.02 M$) before computing feasible ranges. Here, we do not impose any upper or lower bounds on intracellular concentrations at the outset, and all the feasible ranges computed in our analysis arise from the ABCs alone.

3. Given point 2, above, the ABC-based analysis, like TMFA, does not provide the means for prediction of concentrations – particularly given the order or magnitude differences in the ranges that the concentration of a particular metabolite can take (Fig. 2). The ABC-based analysis may result in more restricted feasible concentration ranges. Here, it would be better to quantify the reduction in predicted feasible concentration ranges between TMFA and ABC-based analysis (Fig. 3). In addition, it would be good to quantify the Pearson correlation with TMFA constraints when concentrations are fitted – for comparison. At this point, I wonder if the additional two irreversible reactions found by the ABC-based analysis are due to artefacts mentioned in the points 6 – 9, below.

We computed the degree to which the additional constraints of the ABC-based analysis could reduce the feasible ranges of concentrations and reaction Gibbs energies for the results shown in Fig. 3A,B. The upper bounds on metabolite concentrations and reaction Gibbs energies from TMFA were on average 728% and 11% larger than those from the ABC-based analysis, and the lower bounds were on average 7% and 87% smaller. We have added a paragraph highlighting these results (highlighted on pages 9,10). We also computed the Pearson correlation when only thermodynamic constraints are accounted for. As expected, the closest point inside the CSS defined by thermodynamic constraints alone was better correlated with experimental data compared to the CSS defined by all the ABCs. This is also a consequence of the ABC-based analysis being more restrictive than TMFA because there is more freedom in the feasible set of the optimization problem in which to search for the best fit to experimental data. We have added a figure (Fig. S12) to show these results and a section (Methods: Consistency of the ABCs with Metabolomic Data) at the end of the Method section to discuss this point (highlighted on page 53).

We do not believe that the results in Fig.3A,B are artifacts. Please note that the focus in this paper is on intracellular concentrations. As such, our goal is to examine the phenotypic consequences of imposing the additional constraints of the ABC-based analysis for a given flux state. Granted, the ultimate goal is to integrate the ABC-based analysis with constraint-based models of flux and expression to predict the functional states of the cell under a variety of growth and stress conditions. Accordingly, all the comparative studies between the ABC-based analysis and TMFA that we presented in this paper were performed under the same conditions. For example, for the results in Fig.3B, all the feasible ranges of reaction Gibbs energies were computed for the same flux state. Specifically, the flux directions were fixed for the ABC-based analysis and TMFA. Similarly, the confidence intervals of reaction Gibbs energies were not accounted for in the ABC-based analysis and TMFA. Of course, allowing all the reactions to be reversible and accounting for confidence intervals would render the feasible ranges provided by both the ABC-based analysis and TMFA less restrictive.

We should also emphasize that, in NET or TMFA, upper and lower concentration bounds are provided at the beginning ($10^{-5} M \leq C \leq 0.02 M$) before computing feasible ranges. Here, we do not impose any upper or lower bounds on intracellular concentrations at the outset, and all the feasible ranges computed in our analysis arise from the ABCs alone. For example, for the results in Fig.3B, computations

for the ABC-analysis were performed in the range $6.37 \times 10^{-10} M \leq C \leq \infty M$ (the lower bound corresponds to 1 molecule per cell) and for TMFA in the range $6.37 \times 10^{-10} M \leq C \leq 0.02 M$. In this sense, the initial concentration bounds used for TMFA was more restrictive at the outset than for the ABC-based analysis.

4. The ABC-based approach is applied to a core model of *E. coli* consisting of 78 reactions, 72 cytosolic and 11 periplasmic metabolites. The reason for selecting this model is that the considered metabolites already cover 90% of the observed metabolome (by mole). It is best to be realistic and specify that the approach is applied to a small network due to the computational complexity of the programs that are formulated and some of the developed approaches.

As the reviewer pointed out, we studied a reduced network due to the computational challenges of characterizing the CSS for large, genome scale models. We have added a sentence to section 2.1 to clarify this point (highlighted on page 3). However, it is also important how one might simplify the full metabolic network of an organism without losing too much information about the biophysical underpinnings of cellular metabolism. In the context of constraint-based modeling of intracellular concentrations, considering a subnetworks that includes the most dominant metabolites according to metabolomic data was a natural choice that we adopted in this study.

5. A sampling approach was developed to obtain estimates for the expected concentrations of metabolites. The sampling approach (Eqs. (85) – (86)) relies on generation of random curves in a embedding of a manifold generated by the equalities in the ABC-based analysis. Interestingly, in a recently published companion work by Akbari & Palsson (2020), the proposed sampling procedure failed to identify the feasible concentration region a pathway, like glycolysis, which is considerably smaller than the network analysed here. It would be interesting to show if and to what extent the derived estimates for the expected concentrations are biased.

In our earlier paper that the reviewer is referring to, we examined three approaches to determine the feasibility of the CSS: (i) polynomial optimization, (ii) random sampling, and (iii) global optimization. From these methods, only the global optimization approach could be applied to larger scale problems. In the random sampling approach, we determined the volume of the CSS using acceptance-and-rejection algorithms, which failed in higher dimensional problems. In the global optimization approach, however, we proposed another technique that could handle larger scale problems, which is the one we also propose in the section Methods: Characterization of Concentration Solution Space of the present manuscript. Please note that, this sampling technique requires an interior point at the beginning, from which random curves are generated. We obtain this point by solving Eq. (85), which is a global-optimization problem. In this technique, if the CSS is a disconnected set, then expected concentrations represent part of the CSS, in which the initial interior point falls. In this sense, the expected values may be biased.

Given the nonlinearities of the ABCs, determining whether the CSS is disconnected and, and if so, characterizing the individual parts in a rigorous way is very challenging. For this reason, we proposed Eq. (85), which provides a single-parameter family of interior points, allowing us to perturb the initial point and check whether the initial point falls in another part of the CSS with significantly different characteristics that would affect expected concentrations. For Fig. 2B and Fig. 2C, we performed the computations of expected concentrations for several cases with different interior points by changing the

value of the parameter in the range $w = 0.05 - 0.3$ when solving Eq. (85) and observed no significant difference.

6. The work also relies on determining a flux state and respective inputs. To this end, fluxes were fitted by a bilevel optimization problem (Eq. (93)). The experimentally determined fluxes come with confidence intervals which should be factored in the flux fitting (e.g. via Mahalanobis distance) for unbiased characterization of flux space.

Thank you for this suggestion. We have performed flux computations with and without accounting for the confidence intervals reported by Gerosa et al. for all four carbon sources and compared the results. Overall, incorporating the confidence intervals into computations did not result in a significant difference flux difference or flux-direction changes. Nevertheless, the flux distribution provided by the bilevel optimization problem, for the case where confidence intervals were taken into account, agreed better with the flux data in general (except when pyruvate was the carbon source). We have summarized these results in an excel sheet (Supplementary Data 5.xlsx), which we have submitted in a supplementary file. We also modified Eq. (93) accordingly (highlighted on page 50).

7. Related to point 6, the bilevel program provides a flux estimate which minimizes the second norm of the flux distribution. Are any of the fluxes estimated in such a way zero? If so, based on Eqs. (50) – (52), some of the concentrations of the substrates must be zero or the enzyme should be either not expressed or inactive. Further elaboration of the flux state is needed to justify its usage in the downstream analyses.

The bilevel problem in Eq. (93), like FBA, furnishes zero flux for osmoprotectant transporters/biosynthetic pathways. Of course, these reactions are not essential for growth when the cell is under no stress. These pathways and their respective metabolites were included in our analysis for their role in osmotic stress response. As the reviewer pointed out, a zero flux in the solution of FBA or parsimonious FBA generally implies that the respective enzyme is not expressed or inactive. However, it does not imply that the concentration of the metabolites that determine the rate of that reaction must be zero.

We chose the reduced network in our analysis to account for all the pathways that connect the metabolites whose concentrations were dominant in the metabolome (according to available metabolomic data) and were carrying a flux. We also included some additional pathways that were relevant to stress conditions that we were interested in. Accounting for the thermodynamic constraints of pathways that carry zero flux according to the FBA or parsimonious FBA solutions does not necessarily lead to wrong or biologically irrelevant interpretations. Taking osmoprotectant importers/biosynthetic pathways as an example, these reactions are activated when the cell is perturbed from its optimal growth conditions, in which case all the fluxes, metabolite concentrations, and protein expressions must be adjusted to resume homeostasis under the new conditions. By requiring the concentrations to satisfy the thermodynamic constraints of these additional reactions, we ensure that these reactions can proceed without having to overcome any energy barrier if necessary. Accordingly, we can identify the concentration ranges in which the cell can maintain an optimal growth rate when it is under no stress and can also robustly adjust to new conditions when it is subjected to environmental stresses. A similar line of reasoning was given by Park et al. [2] to explain why glycolytic reactions remain near equilibrium when the cells grow on glucose under no stress.

8. Related to point 6 and 7, the reaction direction is fixed to the net flux in the flux from the bilevel optimization. Does the bilevel optimization not result in alternative solutions? If there is any alternative solution, i.e. flux distribution with different sign pattern, the downstream analysis and resulting findings will be undoubtedly biased. Further support for the decision to fix reaction directionality is needed.

In the bilevel optimization problem Eq. (93), the inner problem is a strictly convex parametric quadratic program (QP). Therefore, it has a unique solution at any given θ . Moreover, the solution $v_1^*(\theta)$ is a continuous and affine function of θ (Theorem 2 of [3]). The outer problem is also a strictly convex QP in v . Let $f(v) := \|v - v^{ex}\|_2$ denote the objective function of this optimization problem. Then, the outer problem can be viewed as a minimization of a composite function with respect to θ , that is $f(\theta) = \|v_1^*(\theta) - v^{ex}\|_2$. Here, $f(\theta)$ is also a convex function with respect to θ because the composition of a convex function with an affine function preserves convexity [4]. Therefore, the final solution of Eq. (93) is unique.

As we stated in response to point 3, the goal in this paper is to identify feasible ranges of metabolite concentrations and reaction Gibbs energies for a given flux state. The flux states we obtained from Eq. (93) for each carbon source match their canonical directions according to ¹³C tracing data. We believe it is unlikely that the reactions involved in glycolysis, TCA cycle, ATPsynthase, nutrient transporters, and ETC could operate in the opposite direction of what we computed for any given carbon source because of how we formulated Eq. (93) in a way that could affect our findings.

9. Similar issues due to alternative optima / concentrations closest to measured ones in Gerosa et al. also bring into question the conclusions from the thermodynamic bottleneck analysis.

Similarly to the flux state, we determined the closest concentrations to measured values by minimizing a second-norm function, which is strictly convex and has a unique solution as a result.

10. How sensitive are the findings in Fig. 3 C based on the imposed constraints on the periplasmic concentration of phosphate? Where do these numbers come from?

We have performed parameter-sweep computations to determine the upper/lower concentration bounds on the cytoplasmic phosphate $[[pi]]_c$ as functions of the periplasmic phosphate concentration $[[pi]]_p$ for the Pit and Pst systems instead of just showing the feasible range at two representative $[[pi]]_p$, as in Fig.3C. We have replaced Fig. S6 with another diagram (on page 51 of the revised manuscript), showing the results of these parameter-sweep computations. This figure now shows the global variation of the feasible range of intracellular phosphate concentration with respect to its periplasmic concentration when either the Pit or Pst system is active. Sensitivities can be ascertained from these global results. The feasible ranges shown in Fig.3C were determined in the same way as those in Fig.2B. All the concentration bounds in this figure were determined for a network, in which the Pit system was the only phosphate transporter. The feasible range of $[[pi]]_c$ shown in Fig.2B is, therefore, identical to that in Fig.3C for the Pit system. To determine the feasible range of $[[pi]]_c$ for the Pst system in Fig.3C, we replaced the Pit transporter with the Pst transporter and performed the same type of computations as before.

11. To make sense of the claims in the transcriptional response to osmotic and acid stress, it is important to show that there are statistically significant association between up/down-regulation of transcripts and changes on the level of flux. Currently, the last two sections read like a series of

observations which are linked, to some extent, to previously published results, but do not provide direct mechanistic explanation for the transcriptional changes.

Decade long efforts to reveal transcriptional regulatory networks have not succeeded yet to give us a comprehensive view of how bacteria regulate their key functions. For instance, the key transcriptional regulators for the translation apparatus remain elusive. Bottom-up detailed molecular mechanistic basis for key transcriptional events in bacteria is still lacking. However, top-down systems level approaches through the use of transcriptional regulatory data are merging. In particular, the use of source signal extraction algorithms (such as ICA) has succeeded in revealing network level mechanisms in the form of sets of independently modulated groups of genes (called iModulons [15], see iModulonDB.org). In particular, we now have a series of stress related iModulons that provide a basis for interpreting the computations that the ABCs enable. Thus, at this point in time, we can only state consistency between computational results and overall transcriptional responses. For example, our study can report that, of the 14 reactions with the highest hydrogen consumption/production in the reduced network, 11 are associated with genes that are differentially expressed in response to acid stress (based on fundamental physico-chemical constraints). This result represents the state of the art in elucidating the consequences of these constraints on evolved stress adaptations responses. Clearly, the reviewer is correct that detailed mechanistic relationships are desirable, but they are unfortunately not achievable at this point in time. However, it seems that the computational and data analytic tools that are now available will be able to bridge this gap in the coming years.

12. The findings are based on transformed Gibbs free energy of reactions without magnesium-bound states, with the reasoning that these will be erroneous due to the usage of reference formation energies from the group contribution method. The group contribution method also provides confidence intervals for the estimates, which are usually incorporated in TMFA. The formulation of the ABC-based analysis should be updated to consider these uncertainties and re-evaluate the findings.

We extracted the confidence intervals for the standard transformed Gibbs energy of reactions reported for group contribution methods, incorporating them into the formulation of the thermodynamic constraint given by Eq. (31). We repeated the computations of the feasible ranges reported in Figs. 2 and 3 with these confidence intervals and found that they did not significantly affect the upper/lower bounds furnished by the ABC-based analysis nor did they alter the general phenotypic consequences of the ABCs that we discussed in the paper. We also compared the extent, to which the inclusion of confidence intervals could relax the upper and lower bounds provided by the ABC-based analysis and TMFA and found that those provided by TMFA were more sensitive to confidence intervals than those by the ABC-based analysis. We have added a section (Methods: Confidence Intervals for Gibbs Energy of Reactions), a figure (Fig. S13), and a table (Table S4) in the revised manuscript, summarizing these findings (highlighted on pages 52 and 54). We have also updated the supplementary package containing our codes and included new subroutines to perform optimizations with confidence intervals.

13. It would be illuminating to specify the reasons why Eqs. (47) – (48) are more computationally tractable than Eqs. (43) and (46).

In the original draft, we had briefly discussed the reasons for why performing certain computations were more tractable in the Y space than the X space in the paragraph preceding Eqs. (47) – (48) and also in Methods: Characterization of Concentration Solution Space. We have added more details in the revised manuscript to clarify these reasons (highlighted on page 36).

Minor

It is essential to specify that the transformed Gibbs free energy is used since pH is fixed a priori to the simulations.

We have corrected all the inconsistencies concerning the terminology of reaction energies and formation energies. Throughout the revised manuscript, $\Delta_r G'^0$ and $\Delta_f G'^0$ are consistently referred to as standard transformed Gibbs energy of reaction and standard transformed Gibbs energy of formation.

References:

- [1] Macklin, D.N., Ahn-Horst, T.A., Choi, H., Ruggero, N.A., Carrera, J., Mason, J.C., Sun, G., Agmon, E., DeFelice, M.M., Maayan, I. and Lane, K., 2020. Simultaneous cross-evaluation of heterogeneous E. coli datasets via mechanistic simulation. *Science*, 369 (6502).
- [2] Park, J.O., Tanner, L.B., Wei, M.H., Khana, D.B., Jacobson, T.B., Zhang, Z., Rubin, S.A., Hsin-Jung Li, S., Higgins, M.B., Stevenson, D.M., Amador-Noguez, D., Rabinowitz, J.D.. Near-equilibrium glycolysis supports metabolic homeostasis and energy yield. *Nat Chem Biol*, 15(10):1001–1008 (2019).
- [3] Bemporad, A., Morari, M., Dua, V., Pistikopoulos, E.N.: The explicit linear quadratic regulator for constrained systems. *Automatica* 38(1), 3–20 (2002).
- [4] Boyd, S., Vandenberghe, L.: *Convex Optimization*. Cambridge University Press, Cambridge (2004).
- [5] Sastry, A.V., Gao, Y., Szubin, R., Hefner, Y., Xu, S., Kim, D., Choudhary, K.S., Yang, L., King, Z.A., Palsson, B.O. The Escherichia coli transcriptome mostly consists of independently regulated modules. *Nat. Comm.* 2019;10(1):1-14.

Reviewers' Comments:

Reviewer #1:

Remarks to the Author:

The authors have improved the manuscript significantly. While I am not satisfied with some of their responses, this dissatisfaction concerns solely the clarity of the presentation, not the validity of the results. To make my new comments understandable without referring back to other documents, I also copied the original comment and the authors' response.

ORIGINAL COMMENT 5. [...] L.182 "(i) They are consistent with previous reports based on 2H and 13C metabolic flux analysis for glycolysis [20] and TCA cycle [21, 22]."). However, no details are given. What is meant by consistency here, and how was it quantified?

RESPONSE: We used two metrics to quantify the consistency between the ABCs and experimental data, namely, Pearson's correlation coefficient R and root mean square deviation RMSD in Fig. 1A. [...]

NEW COMMENT: Presumably the authors mean Fig. 2a, not 1a. However, this figure shows measured *concentrations*, not fluxes, so R and RMSD do not tell us about consistency with previous metabolic flux analyses. Maybe the authors' response to my previous Comment 22 is of relevance here: "Here, consistency with isotope-labeling data refers to the fact that the near equilibrium reactions in Figs. 2C and 4A agree with those reported in the literature based on isotope-labeling data. The data are not shown here, but this point is discussed in the text." This point is, in fact, not explicitly discussed in the text, but it should be and would then address my previous comment.

ORIGINAL COMMENT 13. L.363 "We identified 41 differentially expressed genes with the same regulatory targets" – how were the targets identified?

RESPONSE: Among the genes associated with the reduced network, those whose expression levels changed by more than 1.4 fold [...]

NEW COMMENT: These are "genes with the same regulatory targets ... as osmotic stress". How do you know about their regulatory targets? Do you mean "We identified 41 differentially expressed genes, so we assumed that these are targets of regulation in response to osmotic stress"? Please clarify the sentence.

ORIGINAL COMMENT 14a. Eq.(29) – why do the authors expect the total molar concentration of cytoplasmic + periplasmic metabolites + ions to be bounded, rather than, e.g., their volume or mass concentration?

RESPONSE: We impose the boundedness of the cytoplasm by Eq. (29). This is the boundedness of the total volume or total mass. [...] Moreover, mass and molar concentrations are related to one another through a finite multiplicative constant (i.e. molecular weight). Therefore, the bounded sum of molar concentrations and mass concentrations are the same concepts. [...]

NEW COMMENT: Different molecule species have different molecular weights. Thus, bounds on summed molar concentrations and on summed mass concentrations are only approximately the same concepts. That should be made explicit (or, better still, corrected by using the molecular weights).

ORIGINAL COMMENT 14b. Eq.(31) – why is this thermodynamic constraint referred to in the text as the "mass action principle" (L.730)?

RESPONSE: [...] In the literature, it is sometimes referred to as the second law constraint or thermodynamic constraint. In this paper, the terms mass-action principle and thermodynamic constraints have been interchangeably used.

NEW COMMENT: I understand the somewhat indirect relationship to the law of mass action, but I could not find the term "mass action principle" in a google search, except in the context of mathematical epidemiology. In contrast, the notion of a "thermodynamic constraint" is well established in constraint-based modeling (e.g., thermodynamics-based metabolic flux analysis). Thus, to clearly convey the nature of the constraint, you should consistently use the latter term, not the former.

Signed Review:
Martin Lercher

Reviewer #2:

Remarks to the Author:

I appreciate the additional analyses that the authors have conducted to address the comments pertaining to the consideration of confidence intervals associated with the estimates of standard Gibbs free energy and to fitting of fluxes (by considering error estimates from Gerosa et al.). In addition, the authors have provided justified response about the lack of mechanistic explanations in the interpretation of transcriptional changes.

Importantly, the response provides more emphasis on the differences between TMFA and the proposed ABC-based analysis. I find that these differences must be prominently stressed in the main article, rather than to be buried in the methods (and the additional supplementary figures, e.g. Fig. S12).

Some important points still remain, which can be readily addressed by streamlining the article along the following points:

1. In the response letter, the authors also stressed that the goal of ABC-based analysis is to provide estimates of concentration ranges. This does not coincide with what it stated in the updated abstract, where the main statement is "we formulate a comprehensive set of ten governing abiotic constraints that define possible quantitative metabolomes", as well as in several places throughout the manuscript. If the main outcome of the ABC-based analysis are concentration ranges, then this should be stated precisely as such. In the same direction, the authors should then also discuss the meaning of concentration range in a possibly -- disconnected -- feasible region, whose characterization is arguably very computationally and mathematically challenging.
2. That said, the caption of Figure 2 still talks about "computed concentrations", which are in fact fitted concentrations, and should be stated as such. The sanity check that these studies provide do not correspond to the justification provided in the response (the study of Macklin et al. Science 2020), since no parameters of an enzyme kinetics are used in the present formulation. I understand that the feasible space of ABC-based analysis is smaller than that of TMFA; however, the large ranged defined by the lower and upper bounds does not render it unlikely to find "matching" concentrations in a 70-dimensional space (as claimed in the response).
3. I disagree with the claim that the determined concentrations and fluxes – over all modelled metabolites and reactions -- are unique. The reason is that not all fluxes and concentrations are measured. Therefore, the strictly convex objective function ensures uniqueness only for the fitted values for the measured fluxes / concentrations, but not for the rest. Hence, fixing the directions for some does not ensure those of the others will be also fixed. Nevertheless, with the size of the network considered and the number of fixed-direction reactions (equally treated in ABC-based analyses and TMFA) would have a strong effect on the findings. This can be mentioned in the methods.

Reviewer #1 (Remarks to the Author):

The authors have improved the manuscript significantly. While I am not satisfied with some of their responses, this dissatisfaction concerns solely the clarity of the presentation, not the validity of the results. To make my new comments understandable without referring back to other documents, I also copied the original comment and the authors' response.

ORIGINAL COMMENT 5. [...] L.182 "(i) They are consistent with previous reports based on 2H and 13C metabolic flux analysis for glycolysis [20] and TCA cycle [21, 22]."). However, no details are given. What is meant by consistency here, and how was it quantified?

RESPONSE: We used two metrics to quantify the consistency between the ABCs and experimental data, namely, Pearson's correlation coefficient R and root mean square deviation RMSD in Fig. 1A. [...]

NEW COMMENT: Presumably the authors mean Fig. 2a, not 1a. However, this figure shows measured *concentrations*, not fluxes, so R and RMSD do not tell us about consistency with previous metabolic flux analyses. Maybe the authors' response to my previous Comment 22 is of relevance here: "Here, consistency with isotope-labeling data refers to the fact that the near equilibrium reactions in Figs. 2C and 4A agree with those reported in the literature based on isotope-labeling data. The data are not shown here, but this point is discussed in the text." This point is, in fact, not explicitly discussed in the text, but it should be and would then address my previous comment.

In previous versions of the manuscript, we briefly discussed the consistency of fluxes with isotope labeling data in the first of the four points mentioned in section 2.3 (bottom of page 9 in the revised manuscript). In the revised manuscript, we have clarified this point by explicitly stating that consistency here refers to the near-equilibrium reactions of glycolysis and TCA cycle (highlighted on page 9 of the revised manuscript). We have also added a few sentences to the caption of Fig.2 to make it clear what we mean by near-equilibrium reactions in Fig.2c (highlighted on page 6 of the revised manuscript). Those reactions for which the Gibbs free energy, which is evaluated at a point inside the CSS that best matches metabolomic data, is smaller than the minimum value of $\Delta_r G' / RT$ shown on the y-axis of Fig.2c are considered near equilibrium (reactions corresponding to green bars with no 'x'). These are the same reactions as those shown in light blue in Fig.4a.

ORIGINAL COMMENT 13. L.363 "We identified 41 differentially expressed genes with the same regulatory targets" – how were the targets identified?

RESPONSE: Among the genes associated with the reduced network, those whose expression levels changed by more than 1.4 fold [...]

NEW COMMENT: These are "genes with the same regulatory targets ... as osmotic stress". How do you know about their regulatory targets? Do you mean "We identified 41 differentially expressed genes, so we assumed that these are targets of regulation in response to osmotic stress"? Please clarify the sentence.

We have rephrased this sentence in the revised manuscript to address this point (highlighted on page 16).

ORIGINAL COMMENT 14a. Eq.(29) – why do the authors expect the total molar concentration of cytoplasmic + periplasmic metabolites + ions to be bounded, rather than, e.g., their volume or mass concentration?

RESPONSE: We impose the boundedness of the cytoplasm by Eq. (29). This is the boundedness of the total volume or total mass. [...] Moreover, mass and molar concentrations are related to one another through a finite multiplicative constant (i.e. molecular weight). Therefore, the bounded sum of molar concentrations and mass concentrations are the same concepts. [...]

NEW COMMENT: Different molecule species have different molecular weights. Thus, bounds on summed molar concentrations and on summed mass concentrations are only approximately the same concepts. That should be made explicit (or, better still, corrected by using the molecular weights).

We agree with the reviewer that treating the total cytoplasmic molar concentration $C_{t,c1}$ and total cytoplasmic mass concentration $C_{t,c2}$ as input parameters and imposing a fixed total molar and mass concentration as a constraint are not the same. However, we did not impose this boundedness constraint by requiring the sum of molar or mass concentrations to be equal to some prescribed value. As previously stated, the total molar concentration $C_{t,c}$ in our model is a variable that is determined as part of the solution and is not a fixed parameter. It is coupled to metabolite concentrations through the osmotic-balance constraint. We could reformulate the boundedness and osmotic-balance constraints with respect to mass concentrations, but the solutions would remain identical for the same set of input parameters. For example, solving Eqs. (78) and (79), would result in the same global upper and lower bounds on metabolite concentrations and Gibbs free energy of reactions shown in Figs. 2b,c.

To clarify this point, let C_1 and C_2 denote the CSS defined by Eq. (42) using molar and mass concentrations, respectively. Let also $C_{t,c1}$ and $C_{t,c2}$ be the respective total cytoplasmic concentrations. There is a unique $C_{t,c1}$ corresponding to any point of C_1 and a unique $C_{t,c2}$ corresponding to any point of C_2 . Moreover, there is a one-to-one correspondence (bijection to be specific) between points of C_1 and C_2 because switching from molar concentrations to mass concentrations is equivalent to rescaling the coordinates with respect to which the concentration space is represented. Therefore, to any $C_{t,c1}$, there is uniquely a corresponding $C_{t,c2}$. If we solved any of the optimization problems discussed in this manuscript with respect to mass concentrations, the numerical values of the concentration vector C_2 in C_2 and $C_{t,c2}$ would be different than C_1 in C_1 and $C_{t,c1}$. However, we could transform C_2 and $C_{t,c2}$ back into C_1 and $C_{t,c1}$ by calculating the corresponding molar concentrations.

To further clarify these points, we should emphasize that all the linear constraints in the ABC-based analysis, such as charge balance, buffer capacity, and osmotic balance are explicitly linked to molar concentrations and not mass concentrations. So, it is natural to express the boundedness constraint Eq. (29) in terms of molar concentrations to obtain a consistent system of equations.

ORIGINAL COMMENT 14b. Eq.(31) – why is this thermodynamic constraint referred to in the text as the “mass action principle” (L.730)?

RESPONSE: [...] In the literature, it is sometimes referred to as the second law constraint or thermodynamic constraint. In this paper, the terms mass-action principle and thermodynamic constraints have been interchangeably used.

NEW COMMENT: I understand the somewhat indirect relationship to the law of mass action, but I could not find the term “mass action principle” in a google search, except in the context of mathematical epidemiology. In contrast, the notion of a “thermodynamic constraint” is well established in constraint-based modeling (e.g., thermodynamics-based metabolic flux analysis). Thus, to clearly convey the nature of the constraint, you should consistently use the latter term, not the former.

We have replaced the term ‘mass-action principle’ with ‘the law of mass action’ in the revised manuscript to avoid confusion (highlighted on page 30). We consistently refer to this constraint as thermodynamic constraint in the main text. However, only in the section Methods: Abiotic Constraint, we use the term ‘the law of mass action’ to emphasize that it is not equivalent to the second law, although it is derived from it. We have also added a sentence to this section to clarify that the law of mass action is referred to as thermodynamic constraints in the main text (highlighted on page 30).

Signed Review:

Martin Lercher

Reviewer #2 (Remarks to the Author):

I appreciate the additional analyses that the authors have conducted to address the comments pertaining to the consideration of confidence intervals associated with the estimates of standard Gibbs free energy and to fitting of fluxes (by considering error estimates from Gerosa et al.). In addition, the authors have provided justified response about the lack of mechanistic explanations in the interpretation of transcriptional changes.

Importantly, the response provides more emphasis on the differences between TMFA and the proposed ABC-based analysis. I find that these differences must be prominently stressed in the main article, rather than to be buried in the methods (and the additional supplementary figures, e.g. Fig. S12).

We have added a segment to Section 2.2 of the revised manuscript to address this point (highlighted on page 5).

Some important points still remain, which can be readily addressed by streamlining the article along the following points:

1. In the response letter, the authors also stressed that the goal of ABC-based analysis is to provide estimates of concentration ranges. This does not coincide with what it stated in the updated abstract, where the main statement is “we formulate a comprehensive set of ten governing abiotic constraints that define possible quantitative metabolomes”, as well as in several places throughout the manuscript. If the main outcome of the ABC-based analysis are concentration ranges, then this should be stated precisely as such. In the same direction, the authors should then also discuss the meaning of concentration range in a possibly -- disconnected -- feasible region, whose characterization is arguably very computationally and mathematically challenging.

We believe that there is no inconsistency between this statement in the abstract and the goal of the ABC-based analysis being to provide concentration bounds. Here, quantitative metabolomes refer to the points inside the CSS, which is defined/bounded by the ABCs. Within the framework of the ABC-based analysis, all these metabolomes are feasible. However, to specify which one describes the concentration state of the cell under a particular growth or stress condition, we require additional information. We have added a sentence to Section 2.1 of the revised manuscript to make this point clearer (highlighted on page 3).

We have revised the manuscript to make sure that there are no statements, implying that the ABC-based analysis explicitly predicts the concentration state of the cell. We found one such sentence in the last paragraph of introduction, which we have corrected in the revised manuscript (highlighted on page 2).

We have also added a segment to the section Methods: Characterization of Concentration Solution Space to clarify the interpretation of feasible concentration ranges for concentration solution spaces that are nonconvex and disconnected (highlighted on page 45).

2. That said, the caption of Figure 2 still talks about “computed concentrations”, which are in fact fitted concentrations, and should be stated as such. The sanity check that these studies provide do not correspond to the justification provided in the response (the study of Macklin et al. Science 2020), since no parameters of an enzyme kinetics are used in the present formulation. I understand that the feasible space of ABC-based analysis is smaller than that of TMFA; however, the large range defined by the lower and upper bounds does not render it unlikely to find “matching” concentrations in a 70-dimensional space (as claimed in the response).

In Section 2.1 and the caption of Fig.2, the term ‘computed concentrations’ refers to fitted concentrations. In both cases, what was meant by ‘computed concentrations’ was clarified in the sentence immediately following the first place C_{cm} was mentioned in previous versions of the manuscript. Nevertheless, to clarify this point, we rephrased these sentences in the revised version of the manuscript.

Another point we should emphasize is that even though the feasible ranges appear to be large in Fig. 2, as we pointed out in response to the previous point, it does not imply that the concentration of individual metabolite can freely vary within the respective feasible range independently of other metabolites. This is because the CSS lies at the intersection of a nonconvex set of thermodynamically feasible concentration set and an affine space associated with the equality constraints. Thus, the feasible ranges can be viewed approximately as projections of this nonconvex and possibly disconnected set onto the axis of the coordinate system with respect to which the concentration space is represented. As a result, concentrations can vary within their respective feasible range only if they satisfy all the equality constraints.

We pointed out the work of Macklin et al. in our previous response as an example of a common issue in quantitative models of biological systems, namely relying on parameters that are estimated from heterogeneous datasets generated using different experimental procedures in vitro under a variety of conditions, which might not necessarily be consistent with one another. In the work of Macklin et al., the discrepancy between model predictions and key experimental observations was attributed to inconsistencies among datasets of different type, including kinetic parameters. This issue could also

arise in other complex modeling frameworks that rely on parameters from disparate datasets. We believe that this issue is relevant to the ABC-based analysis that relies on parameters estimated from different data types and various experimental procedures, such as equilibrium data, metabolomic data, hydrogen and metal-ion binding constants, buffer capacity, osmotic pressure, and membrane potential. Therefore, it is important to ensure that the ABC-based analysis is inherently self-consistent, and the various sets of parameters it relies on do not lead to conflicting results.

3. I disagree with the claim that the determined concentrations and fluxes – over all modelled metabolites and reactions -- are unique. The reason is that not all fluxes and concentrations are measured. Therefore, the strictly convex objective function ensures uniqueness only for the fitted values for the measured fluxes / concentrations, but not for the rest. Hence, fixing the directions for some does not ensure those of the others will be also fixed. Nevertheless, with the size of the network considered and the number of fixed-direction reactions (equally treated in ABC-based analyses and TMFA) would have a strong effect on the findings. This can be mentioned in the methods.

We have added a paragraph to the section **Methods: Flux Distribution** in the revised manuscript to address this point (highlighted on page 51).